# Pain-causing stinging nettle toxins target TMEM233 to modulate Na$_V$1.7 function

Sina Jami[1,14], Jennifer R. Deuis[1,14], Tabea Klasfauseweh [1,14], Xiaoyang Cheng[2,3], Sergey Kurdyukov[4], Felicity Chung[4], Andrei L. Okorokov[5], Shengnan Li[5], Jiangtao Zhang[6], Ben Cristofori-Armstrong [7,8], Mathilde R. Israel[9], Robert J. Ju[1], Samuel D. Robinson[1], Peng Zhao[2,3], Lotten Ragnarsson [1], Åsa Andersson[1], Poanna Tran[1], Vanessa Schendel [1], Kirsten L. McMahon[1], Hue N. T. Tran[1], Yanni K.-Y. Chin[7], Yifei Zhu[7], Junyu Liu [7], Theo Crawford [7], Saipriyaa Purushothamvasan[7], Abdella M. Habib[10], David A. Andersson[9], Lachlan D. Rash [8], John N. Wood[5], Jing Zhao [5], Samantha J. Stehbens[1], Mehdi Mobli [7], Andreas Leffler [11], Daohua Jiang [6], James J. Cox [5], Stephen G. Waxman [2,3], Sulayman D. Dib-Hajj [2,3], G. Gregory Neely [4], Thomas Durek [1,12] ✉ & Irina Vetter [1,13] ✉

Voltage-gated sodium (Na$_V$) channels are critical regulators of neuronal excitability and are targeted by many toxins that directly interact with the pore-forming α subunit, typically via extracellular loops of the voltage-sensing domains, or residues forming part of the pore domain. Excelsatoxin A (ExTxA), a pain-causing knottin peptide from the Australian stinging tree *Dendrocnide excelsa*, is the first reported plant-derived Na$_V$ channel modulating peptide toxin. Here we show that TMEM233, a member of the dispanin family of transmembrane proteins expressed in sensory neurons, is essential for pharmacological activity of ExTxA at Na$_V$ channels, and that co-expression of TMEM233 modulates the gating properties of Na$_V$1.7. These findings identify TMEM233 as a previously unknown Na$_V$1.7-interacting protein, position TMEM233 and the dispanins as accessory proteins that are indispensable for toxin-mediated effects on Na$_V$ channel gating, and provide important insights into the function of Na$_V$ channels in sensory neurons.

Voltage-gated sodium (Na$_V$) channels are pore-forming transmembrane proteins that are critical for initiating and propagating action potentials in excitable cells. In sensory neurons, at least five of the nine mammalian isoforms Na$_V$1.1-Na$_V$1.9 are expressed, where they contribute to physiological and pathological neuronal excitability[1]. Accordingly, Na$_V$ channels are important pain targets, in particular the tetrodotoxin-sensitive Na$_V$1.7 as well as the tetrodotoxin-resistant Na$_V$1.8 and Na$_V$1.9, whose expression is largely restricted to nociceptors.

Na$_V$ channels are composed of a Na$^+$ ion conducting α subunit that can associate with one or more accessory β subunits, and

respond to changes in transmembrane voltage with conformational rearrangements leading to opening of the channel pore[2–5]. Given their central role in the function of excitable cells across phyla, it is not surprising that numerous toxins have evolved to target Na$_V$ channels[6,7]. These compounds typically directly interact with the α subunit, either at the pore or via extracellular loops of the voltage-sensing domains, and belong to structurally diverse classes that inhibit (e.g. the μ-conotoxins or μ-theraphotoxins) or enhance (e.g. the δ-theraphotoxins, ί-conotoxins, or α-scorpion toxins) channel function[6,7]. These peptides have provided considerable insight into the structure and function of Na$_V$ channels, as well as the role of

specific Na$_V$ channel isoforms in neuronal function and pain signaling.

Recently, we identified a novel family of plant venom-derived Na$_V$ channel-targeting toxins, called gympietides, from members of the *Urticaceae* family native to Australia, including the Australian stinging trees (genus *Dendrocnide*)[8]. These nettles are renowned for inflicting extremely painful stings that are characterized by acute electric shock-like, piercing, pricking and burning sensations lasting for many hours, followed by intermittent painful flares and allodynia that persists for days or even weeks[9]. The gympietides, found in the fluid-filled stinging trichomes covering the leaves and stems of these members of the nettle family, were identified as the main causative agents eliciting spontaneous action potential discharge, an axon reflex flare and nocifensive behaviors in vivo[8]. These remarkable peptides are characterized by a unique primary amino acid sequence and a tertiary structure closely resembling inhibitory cystine knot peptides typically found in animal venoms[8]. Consistent with this high structural homology, synthetic gympietides inhibit Na$_V$ channel inactivation in dissociated dorsal root ganglion (DRG) neurons, analogous to effects caused by Na$_V$ channel-targeting inhibitory cystine knot peptides from cone snail or spider venoms[8]. However, the mechanism by which excelsatoxin A (ExTxA), the first identified gympietide, exerts its effect on Na$_V$ channels, and the specificity of this interaction among members of the Na$_V$ channel family, are not known.

In this study, we sought to investigate the molecular determinants underlying activity of this class of Na$_V$-modulating peptides. We report that ExTxA-induced removal of fast inactivation of Na$_V$ currents is not observed upon application of ExTxA to Na$_V$ α subunits alone or when co-expressed with β subunits. Instead, toxin activity requires co-expression of TMEM233, a poorly described member of the dispanin family that we find is highly expressed in sensory neurons, and that can associate with Na$_V$1.7 to subtly modify inactivation properties. These results provide important insights into the function of Na$_V$ channels in sensory neurons, identify TMEM233 as a previously unknown Na$_V$1.7-interacting protein, and describe the dispanins as bifunctional proteins that cause allosteric changes in channel gating upon binding of pain-causing venom peptides.

## Results

### ExTxA inhibits inactivation of Na$_V$1.7 in sensory neurons but not heterologous expression systems

ExTxA, the first identified member of the gympietide family of stinging nettle toxins, elicits spontaneous pain behaviors and an axon reflex flare following intraplantar administration in mice, similar to the symptomatology experienced by humans following stings by *Dendrocnide excelsa* or *D. moroides*[8]. In DRG neurons, a major effect of ExTxA is a striking inhibition of Na$_V$ inactivation, leading to persistent currents that likely contribute to enhanced excitability and spontaneous action potential firing[8]. We show in this study that we consistently observed this ExTxA-mediated effect on TTX-sensitive currents in DRG neurons (Fig. 1a, b). Smaller, but statistically significant effects were also observed on TTX-resistant currents mediated by Na$_V$1.8 (Fig. 1c, d). However, essentially no effects were observed at the persistent currents known to be mediated by Na$_V$1.9 (Fig. 1e, f). Activity at Na$_V$1.7, the major TTX-s isoform expressed in nociceptors, was evidenced by the effects of ExTxA (100 nM) on Na$_V$ current in human iPSC-derived sensory neurons, where the ExTxA-induced persistent current was largely inhibited by the selective Na$_V$1.7 blocker Pn3a[10] (100 nM) (Fig. 1g, h). ExTxA-induced persistent TTX-sensitive currents in TE-671 neuroblastoma cells were also significantly reduced by Pn3a,

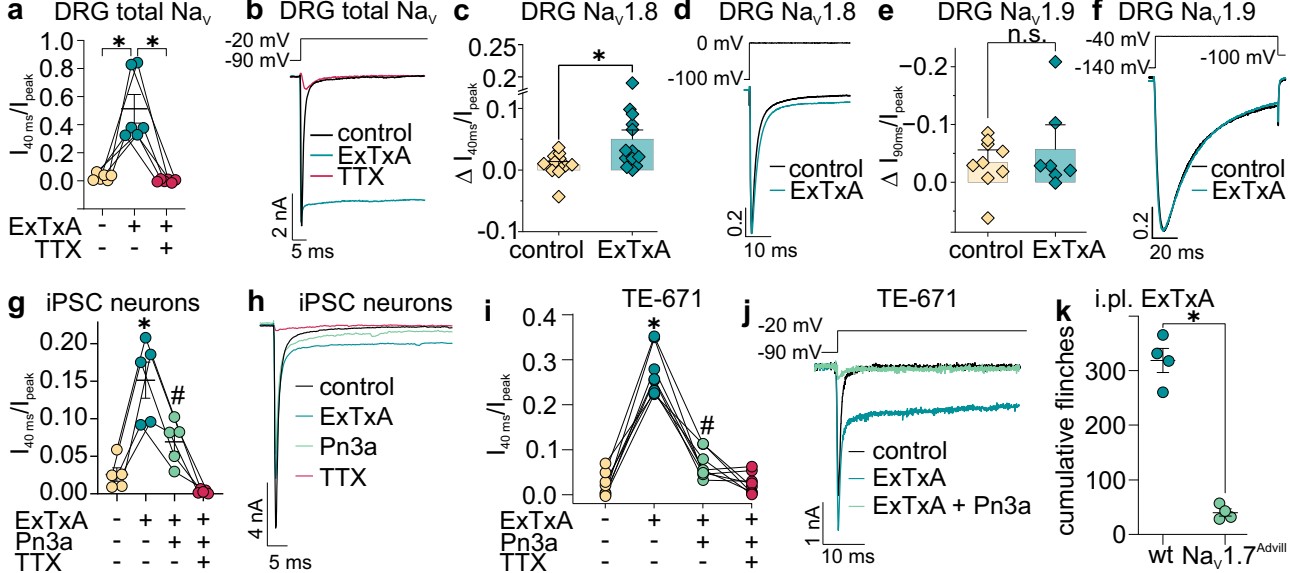

**Fig. 1 | ExTxA inhibits Na$_V$1.7 inactivation in neurons and TE-671 cells. a** Total persistent Na$_V$ current in DRG neurons after perfusion of buffer control (0.1% BSA), ExTxA (100 nM) and ExTxA + TTX (1 μM). ExTxA induces persistent Na$_V$ currents that are blocked by TTX. **b** Representative recording from a DRG neuron showing large TTX-s persistent current induced by ExTxA (100 nM). **c** Change in persistent Na$_V$1.8 current ($\Delta I_{40\,ms}/I_{peak}$; pre- and post-perfusion of buffer control (0.1% BSA) and ExTxA (100 nM)) in DRG neurons from Na$_V$1.9$^{-/-}$ mice in the presence of TTX (1 μM). ExTxA slightly increased persistent currents of Na$_V$1.8 channels. **d** Representative normalized Na$_V$1.8 currents from buffer control (0.1% BSA) and ExTxA (100 nM)-treated Na$_V$1.9$^{-/-}$ DRG neurons. **e** Change in persistent Na$_V$1.9 current ($\Delta I_{90\,ms}/I_{peak}$; pre- and post-perfusion of buffer control (0.1% BSA) and ExTxA (1 μM)) in DRG neurons from Na$_V$1.8$^{-/-}$ mice in the presence of TTX (1 μM). ExTxA did not affect persistent currents of Na$_V$1.9 channels. **f** Representative

normalized Na$_V$1.9 currents of buffer control (0.1% BSA) and ExTxA (1 μM)-treated Na$_V$1.8$^{-/-}$ DRG neurons. **g** Effect of ExTxA (100 nM) on Na$_V$1.7 current in human iPSC-derived sensory neurons. *$p$ < 0.05 (control vs ExTxA); #$p$ < 0.05 (ExTxA vs ExTxA +Pn3a). **h** Representative Na$_V$ current in human iPSC-derived sensory neurons showing effect of ExTxA (100 nM) as well as inhibition by the selective Na$_V$1.7 blocker Pn3a (100 nM), and TTX (1 μM). **i** Effect of ExTxA (100 nM) on persistent current ($I_{40ms}/I_{Peak}$) in TE-671 cells endogenously expressing Pn3a-sensitive Na$_V$1.7. **j** Representative current traces of ExTxA-induced effects on Na$_V$1.7 endogenously expressed in TE-671 cells. **k** Nocifensive behaviors (cumulative paw licks and flinches over 60 min) induced by intraplantar injection of ExTxA (10 nM) in wild-type controls (wt) and Na$_V$1.7$^{Advill}$ mice. *n* values and statistical information are detailed in Supplementary Table 1. Source data are included as a Source Data file.

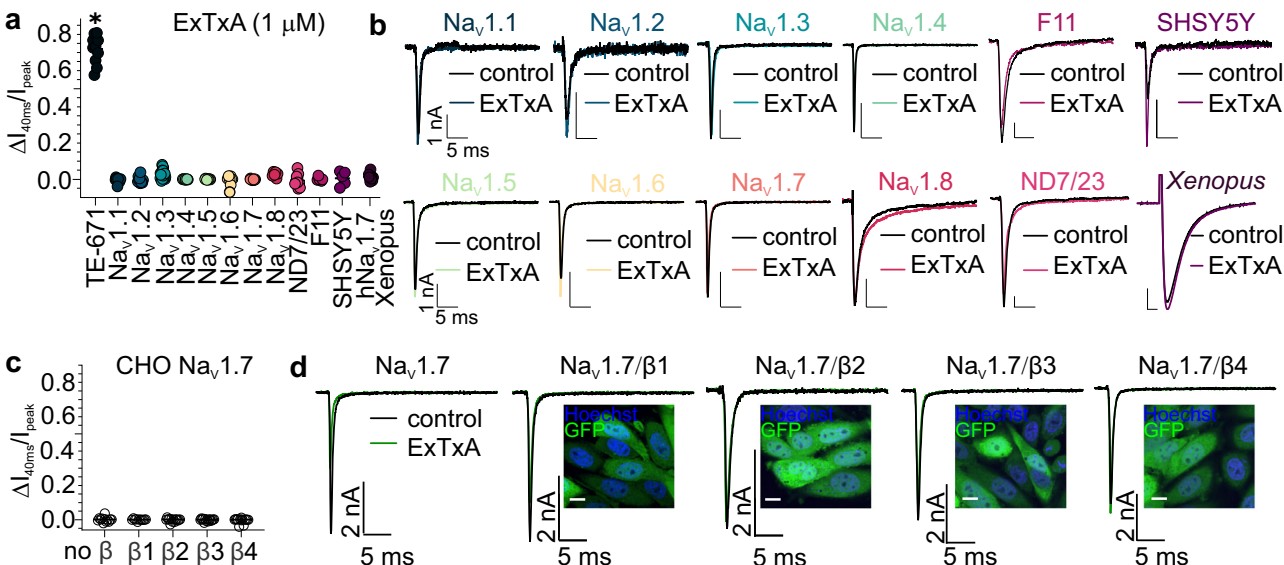

**Fig. 2 | ExTxA does not inhibit Na$_V$1.7 inactivation heterologous expression systems. a** Persistent current ($I_{40\,ms}/I_{peak}$) induced by ExTxA (1 μM) in TE-671 cells, HEK293 cells stably expressing β1 and hNa$_V$1.1, hNa$_V$1.2, hNa$_V$1.3, hNa$_V$1.4, hNa$_V$1.5, hNa$_V$1.6, hNa$_V$1.7 or CHO cells expressing hNa$_V$1.8/β3, ND7/23 cells, F11 cells, SH-SY5Y neuroblastoma cells and Xenopus oocytes expressing hNa$_V$1.7. **b** Representative control (black) and ExTxA-treated (1 μM, colored) normalized current traces of data shown in (**a**). Current was elicited by a depolarizing step to −20 mV (−10 mV for oocytes) from a holding potential of −90 mV. Scale bar: all 1 nA, 5 ms except oocytes (1 μA, 5 ms). **c** ExTxA (1 μM) effect on persistent current ($I_{40\,ms}/I_{peak}$) in CHO cells expressing hNa$_V$1.7 with or without the β1, β2, β3 or β4 subunit. **d** Representative control (black) and ExTxA-treated (1 μM, green) current traces of data shown in (**c**). Inset: Expression of GFP confirming successful transfection of the GFP-IRES β subunit plasmids. Scale bar, 10 μm. Data are shown as mean ± SEM; n.s., not significant; *,#$p < 0.05$. $n$ values and statistical information are detailed in Supplementary Table 1. Source data are included as a Source Data file.

consistent with our previous observation that TE-671 cells express Na$_V$1.7 as the predominant TTX-sensitive Na$_V$ subtype (Fig. 1i, j)[8,11]. Moreover, nocifensive responses induced by intraplantar (i.pl.) injection of ExTxA (10 nM) were abolished in Na$_V$1.7$^{Advill}$ knockout mice (Fig. 1k). This is consistent with our previous observation that Pn3a inhibited ExTxA-induced pain-like responses in mice, and supports the evolution of the gympietides as vertebrate-specific pain-causing defensive agents[8]. To evaluate the activity of ExTxA at invertebrate targets, we assessed the effects of ExTxA in *Drosophila melanogaster*, where injection of toxin (0.5 nmol)−at doses several fold higher than those that induce nocifensive responses in mice−did not elicit paralysis (control 1/25; ExTxA 2/52 after 15 min) or death (control 0/25; ExTxA 1/52 after 2 h), consistent with a lack of insecticidal activity.

In subsequent efforts to further define the selectivity of ExTxA at mammalian Na$_V$ channel subtypes, we were surprised that the effect on Na$_V$ inactivation observed in mouse DRG neurons, human iPSC-derived sensory neurons and TE-671 neuroblastoma cells (Fig. 1a–k, Fig. 2a) could not be reproduced in HEK293 or CHO cells stably expressing hNa$_V$1.1–1.8, where ExTxA (1 μM) failed to induce persistent currents (Fig. 2a, b). ExTxA also had no effect on hNa$_V$1.7 expressed in *Xenopus laevis* oocytes, or on Na$_V$ currents in immortalized neuronal cell lines (F11, ND7/23, SH-SY5Y) known to endogenously express Na$_V$1.7 (Fig. 2a, b). We hypothesized that ExTxA activity might be reliant on auxiliary Na$_V$ subunits expressed in DRG neurons and TE-671 cells, and assessed peptide activity in CHO cells expressing Na$_V$1.7 and the Na$_V$ β1-β4 subunits. However, although successful transfection with β subunits was confirmed by the presence of the transfection marker GFP, ExTxA did not affect inactivation of Na$_V$1.7-mediated currents in the presence of the β1, β2, β3 or β4 subunit (Fig. 2c, d).

## TMEM233 is required for ExTxA-induced inhibition of Na$_V$1.7 inactivation

To identify factors that might mediate ExTxA-sensitivity, we next performed a genome-scale lentivirus-CRISPR knockdown screen in TE-671 cells (Supplementary Fig. S1a), in which ExTxA inhibits the

inactivation of endogenously expressed Na$_V$1.7. Although exposure to ExTxA (1 μM) alone did not cause visible toxicity, co-incubation with non-cytotoxic concentrations of veratridine (5 μM) and ouabain (20 nM) led to ExTxA (1 μM)-mediated cell death (Supplementary Fig. S1b). This cytotoxicity was prevented by the knockdown of *SCN9A* (the gene encoding the Na$_V$1.7 α subunit) as well as ten other genes (*RNF121, GPAA1, PIGT, CRELD1, PIGK, STT3B, TMEM233, PIGS, MMGT1,* and *LMAN2L*), and exacerbated by knockdown of *SPAG5, TMEM161B* and *NEDD4L* (Fig. 3a and Supplementary Fig. S1c). In contrast, treatment with veratridine and ouabain alone did not lead to any sgRNA enrichment or depletion (Supplementary Fig. S1d). Apart from the E3-ubiquitin ligases RNF121 and NEDD4L, which affect degradation and membrane localization of Na$_V$ in vertebrates[12,13], none of these genes have previously been implicated in either toxin binding, or in affecting Na$_V$ function. We thus next assessed whether over-expression of any of these genes could confer ExTxA-sensitivity to HEK293-Na$_V$1.7 cells. Of all 10 gene hits from our TKOv3 lentivirus-CRISPR screen, only expression of TMEM233 imparted ExTxA-sensitivity to Na$_V$1.7-expressing HEK293 cells in a fluorescence membrane potential assay (Fig. 3b), suggesting that TMEM233 is required for ExTxA-induced inhibition of Na$_V$1.7 inactivation. Indeed, co-expression of TMEM233 and Na$_V$1.7 in heterologous expression systems perfectly recapitulated the effects of ExTxA on Na$_V$ current inactivation observed in DRG neurons and TE-671 cells (Fig. 3c and Supplementary Fig. S1e, f). Expression of TMEM233 alone in cells lacking endogenous Na$_V$ currents did not result in either voltage-gated or ExTxA-induced currents (Supplementary Fig. S1g, h). Moreover, over-expression of TMEM233 in HEK293 cells stably expressing Na$_V$1.1-Na$_V$1.6 also resulted in ExTxA-mediated persistent currents (Supplementary Fig. S2a–g).

Conversely, ExTxA-induced effects on Na$_V$ inactivation were abolished following CRISPR/Cas9-mediated TMEM233 knockdown in TE-671 cells (Fig. 3d). Similarly, ExTxA-induced persistent Na$_V$ currents were abrogated in small-diameter nociceptors derived from Tmem233$^{Cre/Cre}$ knockout mice[14] (Fig. 3e). Toxin-induced Ca$^{2+}$-influx in DRG neurons from Tmem233$^{Cre/Cre}$ KO mice was also strongly reduced,

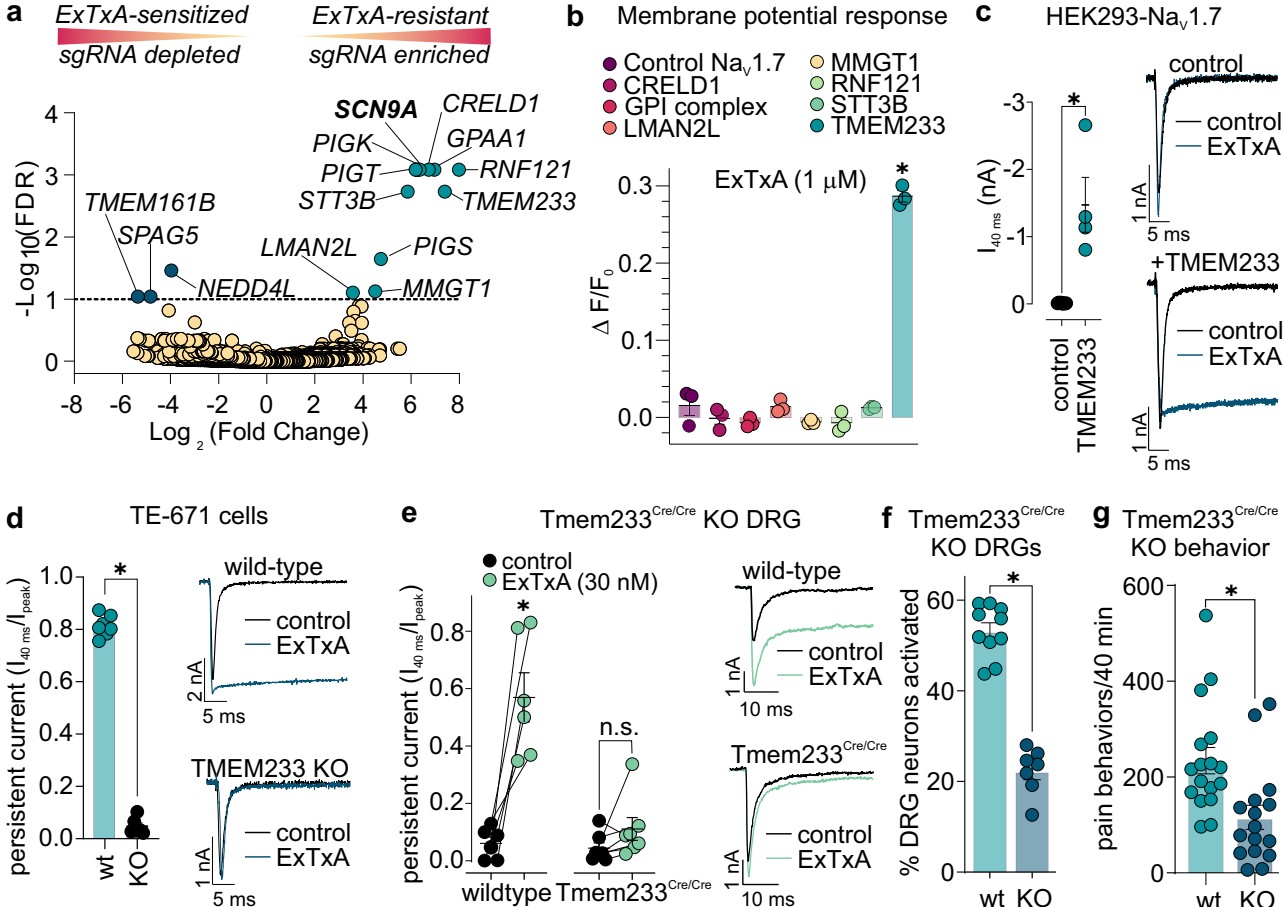

**Fig. 3 | TMEM233 is required for ExTxA-induced inhibition of Na$_V$1.7 inactivation. a** LentiCRISPR-Cas9 TKOv3 screen showing depletion and enrichment of sgRNAs in control (veratridine, 5 μM; ouabain, 20 nM) and ExTxA-treated (ExTxA, 1 μM; veratridine, 5 μM; ouabain, 20 nM) TE-671 cells. **b** ExTxA (1 μM)-induced change of membrane potential dye fluorescence (ΔF/F$_0$) in HEK293-Na$_V$1.7 cells co-transfected with *CRELD1*, GPI complex (*GPAA1, PIGU, PIGS, PIGK, PIGT*), *LMAN2L, MMGT1, RNF121, STT3B*, and *TMEM233*. **c** Persistent currents (I$_{40 \text{ ms}}$; nA) induced by ExTxA (1 μM) in HEK293-Na$_V$1.7 cells (control) and HEK293-Na$_V$1.7 cells co-transfected with TMEM233 (TMEM233). Right panels, representative current traces; depolarization to −20 mV from −90 mV holding potential. **d** Normalized persistent currents (I$_{40 \text{ ms}}$/I$_{\text{peak}}$) induced by ExTxA (1 μM) in wild-type TE-671 cells (wt) and following CRISPR/Cas9-mediated knockdown of TMEM233 (KO). Right panels, representative current traces; depolarization to −20 mV from −90 mV holding potential. **e** Normalized total persistent Na$_V$ current (I$_{40 \text{ ms}}$/I$_{\text{peak}}$) in presence of buffer (0.1% BSA; pre) and after ExTxA (30 nM) in DRG neurons from wild-type and Tmem233$^{\text{Cre/Cre}}$ knockout mice. Right panels, representative current traces; depolarization to −20 mV from −80 mV holding potential. **f** The percentage of cultured DRG neurons from wild-type C57BL/6 (wt) and Tmem233$^{\text{Cre/Cre}}$ knockout (KO) mice activated by ExTxA (20 nM). **g** Cumulative pain behaviors (paw licks and flinches over 40 min) following intraplantar injection of ExTxA in littermate C57BL/6 (wt) and Tmem233$^{\text{Cre/Cre}}$ knockout mice (KO). Data are shown as mean ± SEM; *$p < 0.05$. *n* values and statistical information are detailed in Supplementary Table 1. Source data are provided as a Source Data file.

suggesting that TMEM233 is indeed important for ExTxA-evoked action potential firing in sensory neurons (Fig. 3f). Accordingly, ExTxA-induced nocifensive behaviors were significantly reduced in Tmem233$^{\text{Cre/Cre}}$ KO mice (Fig. 3g).

### ExTxA effects on Na$_V$1.7 are mediated by a direct interaction of the toxin with TMEM233

We next assessed the binding of N-terminally biotinylated or Alexa488-tagged ExTxA, which retained functional activity (Supplementary Fig. S3a–c), to Na$_V$1.7 and TMEM233. Flow cytometry analysis of cells treated with biotin-ExTxA showed a significantly increased signal in TMEM233-expressing HEK293 cells but not in HEK293-Na$_V$1.7 cells compared to control, suggesting that ExTxA directly binds to TMEM233 (Fig. 4a and Supplementary Fig. S3d). Similarly, fluorescence signals from Alexa488-tagged ExTxA were observed only in TMEM233-expressing cells, with the EC$_{50}$ of toxin binding to Na$_V$1.7/TMEM233 co-expressing cells (EC$_{50}$ 125.9 ± 19.2 nM, *n* = 3) being comparable to the functional potency on Na$_V$1.7 inactivation in the same cells (EC$_{50}$ 80.9 ± 29.1 nM, *n* = 7) (Fig. 4b and Supplementary Fig. S3c). The

binding of biotinylated ExTxA to TMEM233 was also confirmed by confocal microscopy, which showed punctate toxin labeling predominantly on the cell surface (Fig. 4c). Moreover, in electrophysiology experiments, ExTxA included in the intracellular solution (ICS) did not affect inactivation of Na$_V$1.7 co-expressed with TMEM233, while toxin in the extracellular solution (ECS) robustly inhibited inactivation (Fig. 4d). Thus, ExTxA appears to interact with TMEM233 from the extracellular side, which is also supported by our observation that ExTxA associates with dodecylphosphocholine micelles (Supplementary Fig. S4a). In addition to profound effects on Na$_V$1.7 inactivation, extracellular exposure to ExTxA also resulted in a prominent (11.6 mV) hyperpolarizing shift in the voltage-dependence of activation, a strong depolarizing shift (18.0 mV) in the voltage-dependence of fast inactivation, a decrease in the time constants of recovery from inactivation and an increase of currents induced by slow voltage-ramps (Supplementary Fig. S4b–g), consistent with the excitatory effect of ExTxA on nociceptors[8].

TMEM233 is a member of the dispanin family, with the closest similarity to PRRT2 and TRARG1 (previously known as TUSC5)[15].

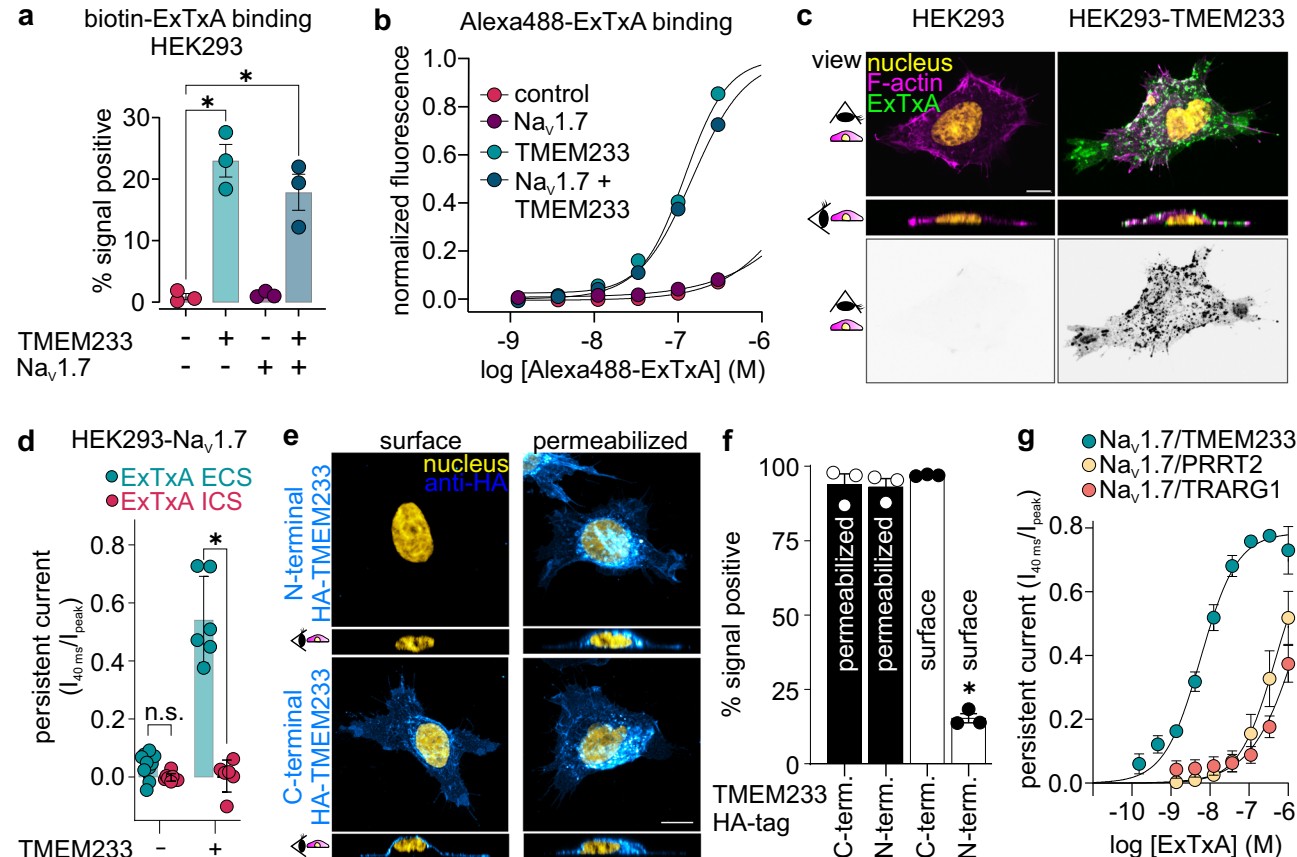

**Fig. 4 | ExTxA interacts directly with TMEM233. a** Percentage of fluorescence signal-positive cells, determined by flow cytometry, following staining with biotinylated ExTxA (1 µM) and DyLight680-conjugated streptavidin in untransfected or TMEM233-transfected HEK293 and HEK293-Na$_V$1.7 cells. **b** Fluorescence intensity following staining with Alexa488-conjugated ExTxA (1–300 nM) in mock- or TMEM233-transfected HEK293 and HEK293-Na$_V$1.7 cells. **c** Confocal microscopy image showing Alexa488-streptavidin signal only in TMEM233-expressing cells following incubation with biotinylated ExTxA (1 µM). Yellow: nucleus; magenta: F-actin; green: ExTxA. Middle panels: orthogonal view. Bottom panels: inverted grayscale of ExTxA signal. Scale bar, 10 µm. Image representative of 3 independent experiments. **d** Normalized persistent current ($I_{40\ ms}/I_{peak}$), induced by ExTxA (100 nM) either in the extracellular solution (ECS) or intracellular solution (ICS), in HEK293-Na$_V$1.7 cells co-transfected with TMEM233. **e** Confocal microscopy image showing anti-HA immunofluorescence in non-permeabilized (surface) or permeabilized HEK293 cells transfected with either C-terminal or N-terminal HA-tagged TMEM233. Blue, anti-HA; yellow, nucleus. Scale bar, 10 µm. **f** Percentage of fluorescence signal-positive cells, determined by flow cytometry, of permeabilized or non-permeabilized HEK293 cells transfected with either C-terminal or N-terminal HA-tagged TMEM233 following staining with anti-HA antibody. **g** Concentration-response curves of ExTxA-induced normalized persistent currents ($I_{40\ ms}/I_{peak}$) in HEK293-Na$_V$1.7 cells co-transfected with TMEM233, PRRT2 or TRARG1. Data are shown as mean ± SEM; *$p < 0.05$. *n* values and statistical information are detailed in Supplementary Table 1. Source data are included as a Source Data file.

Although TMEM233 was originally predicted to comprise two transmembrane helices and extracellular N- and C-termini[15], the paralogs PRRT2 and TRARG1 were recently confirmed to adopt a single transmembrane topology[16,17]. Specifically, in these subfamily B dispanin members the N-terminal domain is localized intracellularly and is followed by a central hydrophobic segment forming a membrane re-entrant loop, and another membrane-spanning segment that positions the short C-terminus extracellularly[16,17]. We sought to determine the membrane topology of TMEM233 and generated C- and N-terminally HA-tagged constructs. Immunofluorescence (Fig. 4e) and flow cytometry (Fig. 4f) experiments in permeabilized and non-permeabilized cells demonstrated that, similar to PRRT2 and TRARG1, TMEM233 also possesses an extracellular C-terminus and an intracellular N-terminus (Fig. 4e, f; Supplementary Fig. S4h). Based on the similar sequence (Supplementary Fig. S4i) and topology, we were curious whether co-expression of PRRT2 or TRARG1 could sensitize Na$_V$1.7 to ExTxA. Indeed, co-expression of PRRT2 or TRARG1 with Na$_V$1.7 resulted in ExTxA-induced persistent currents, albeit the toxin was significantly more potent when co-expressed with TMEM233 (Fig. 4g and Supplementary Fig. S4j–l).

## TMEM233 associates with Na$_V$1.7

Our data indicate that the effects of ExTxA on Na$_V$ function are mediated via a direct interaction of the toxin with TMEM233, implying that TMEM233 is located in close proximity to Na$_V$1.7 in neurons and represents an interacting protein. Although no functional role for TMEM233 has previously been described, TMEM233 is notable for its DRG subtype-specific expression pattern, with single-cell sequencing suggesting that expression is restricted to nociceptors[14,18,19]. Using a Tmem233 Cre-knockin mouse line crossed with a CAG-floxed stop tdTomato reporter line, in which tdTomato is expressed under the *Tmem233* regulatory elements, we found that small diameter, predominantly *Nefh* (neurofilament heavy chain)-negative DRG neurons expressed tdTomato (Fig. 5a and inset). Additionally, RNAscope In Situ Hybridization (ISH) confirmed co-expression of both *Tmem233* and *Scn9a* by some neurons, while large neurons expressing *Nefh* mRNA showed limited signal co-localization with *Scn9a* or *Tmem233* mRNAs (Fig. 5b). As the ExTxA-induced effects on Na$_V$1.7 suggest a functional interaction between Na$_V$1.7 and TMEM233, we next assessed whether the two proteins can associate in the absence of toxin. When co-expressed with GFP- and Twin-Strep-tagged Na$_V$1.7, N-terminally

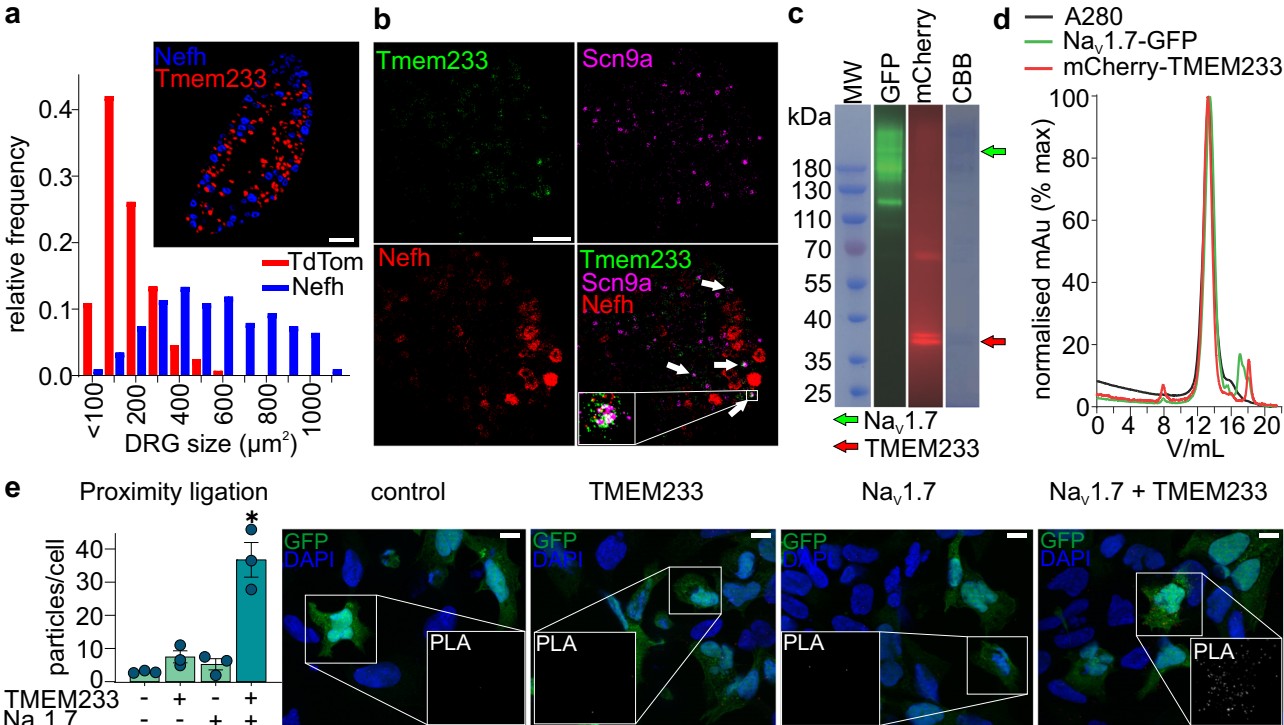

**Fig. 5 | TMEM233 is expressed in dorsal root ganglion neurons and can associate with Na$_V$1.7. a** Expression of tdTomato (red) and *Nefh* RNA, detected with RNAscope analysis (TS405, blue), in lumbar DRGs isolated from adult Tmem233$^{Cre}$/Rosa-CAG-flox-stop-tdTomato mice. Inset, representative image of DRG section used for analysis, from 4 independent experiments. Scale bar, 200 µm. **b** *Tmem233* and *Scn9a* RNA expression and localization in fresh-frozen sections of mouse DRGs determined by RNAscope analysis. Localization of *Tmem233* mRNA (green, AF488) was compared to *Nefh* mRNA (red, Opal570) and *Scn9a* mRNA (far-red, Opal650, shown in magenta). Arrows, neurons expressing both *Tmem233* and *Scn9a* leading to white signal color overlap. Scale bar, 100 µm. Image representative of 2 independent experiments. **c** SDS-PAGE gel of purified Na$_V$1.7-TMEM233 complex. CBB (Coomassie brilliant blue staining), GFP (green fluorescent protein) fluorescence and mCherry fluorescence of the same SDS-PAGE gel. The green and red arrow indicate Na$_V$1.7-GFP and mCherry-TMEM233 bands, respectively. MW, molecular weight marker (kDa). **d** Normalized size-exclusion chromatography profile of the purified Na$_V$1.7-TMEM233 complex. Signals of total protein (black), GFP (green) and mCherry (red) were detected simultaneously. **e** Left: Quantification of proximity ligation assay signal in GFP-positive HEK293 cells transfected with GFP as well as N-terminal FLAG-tagged hNa$_V$1.7 and N-terminal HA-tagged TMEM233 alone or in combination. Right: Representative images showing nuclei (DAPI, blue), GFP (Green) and proximity ligation signal (red channel, enlarged view in inset shown in grayscale for clarity). Scale bar, 10 µm. Data are shown as mean ± SEM; *, p < 0.05. n values and statistical information are detailed in Supplementary Table 1. Source data are included as a Source Data file.

mCherry-tagged TMEM233 could indeed be co-purified together with Na$_V$1.7, suggesting that Na$_V$1.7 and TMEM233 can form a complex (Fig. 5c, d). In addition, using proximity-ligation assays we observed signal amplification only in cells co-expressing N-terminally FLAG-tagged Na$_V$1.7 and N-terminally HA-tagged TMEM233, indicating that the two proteins are present in close proximity to each other (Fig. 5e)[20].

## Co-expression of TMEM233 affects fast and slow inactivation of Na$_V$1.7

As the ExTxA-induced effects on Na$_V$1.7 suggest a functional interaction between Na$_V$1.7 and TMEM233, we next evaluated the effects of TMEM233 co-expression on the biophysical properties of Na$_V$1.7. Compared to mock-transfected HEK293-Na$_V$1.7 cells, co-expression of TMEM233 had minor, though statistically significant, effects on the voltage-dependence of fast inactivation (−3.9 mV), but no effect on the voltage-dependence of activation (Fig. 6a–e). Small effects on the voltage-dependence of slow inactivation (−5.5 mV; Fig. 6f, g) as well as the time constant of recovery from fast inactivation (Fig. 6h, i) were also observed. Given these effects on inactivation parameters, we next tested whether TMEM233 might also affect Na$_V$1.7-mediated ramp currents or use-dependence of Na$_V$1.7. However, co-expression of TMEM233 did not affect ramp currents, and only minor effects on use dependence were observed at the highest tested frequency (20 Hz) (Fig. 6j–l). Co-expression of TMEM233 also had similar minor effects on Na$_V$1.7 function in *Xenopus oocytes*, most notably a small but

significant effect on recovery from fast inactivation and a small shift in the voltage-dependence of slow inactivation (Supplementary Fig. S5a−e). We hypothesized that these effects may be mediated by the intracellular N-terminal domain of TMEM233, and assessed the effects of an N-terminally truncated TMEM233 mutant on recovery from fast inactivation. Indeed, co-expression of Na$_V$1.7 with TMEM233 lacking the first 34 N-terminal residues no longer affected recovery from fast inactivation (Fig. 6m), although ExTxA was still able to inhibit fast inactivation (Supplementary Fig. S5f). Conversely, the inclusion of a synthetic peptide corresponding to the TMEM233 N-terminus in the intracellular solution in HEK293-Na$_V$1.7 cells recapitulated the effects observed with co-expression of full-length TMEM233 on Na$_V$1.7 recovery from fast inactivation, suggesting that this effect is mediated via interactions with the TMEM233 N-terminus (Fig. 6m). Together, our data indicate that TMEM233 is a previously unknown Na$_V$1.7-interacting protein targeted by pain-causing venom peptides from Australian stinging nettles, providing insight into the Na$_V$1.7 interactome that may ultimately be amenable to pharmacological or gene therapy modulation for therapeutic benefit.

## Discussion

The gympietides are a novel class of Na$_V$-targeting peptide toxins notable for several distinctive properties, including a unique primary structure sharing no significant similarity to known sequences apart from plant-derived albumins at approximately 50% identity[8]. The

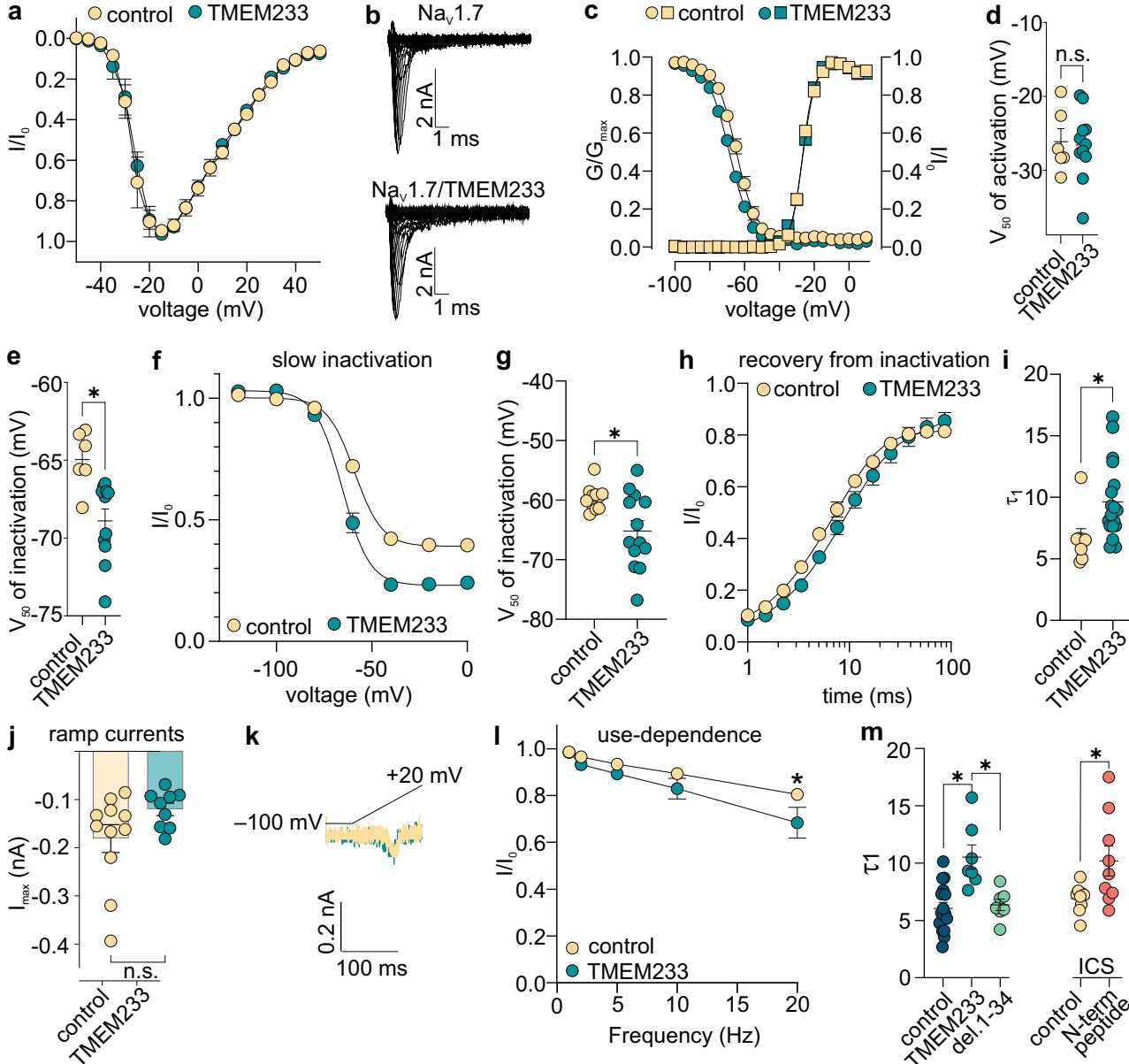

**Fig. 6 | Effects of TMEM233 on Na$_V$1.7 function. a** Current-voltage relationship and (**b**) family of sample traces from hNa$_V$1.7 ($n = 6$) co-expressed with TMEM233 ($n = 11$) in HEK293 cells. **c** Superimposed conductance–voltage (G–V, squares) and steady-state fast inactivation (circles) curves, of mock- (control, yellow) and TMEM233-transfected (teal) HEK293-Na$_V$1.7 cells. **d** V$_{50}$ of activation and (**e**) V$_{50}$ of inactivation of mock- and TMEM233-transfected HEK293-Na$_V$1.7 cells. **f** Voltage-dependence of slow inactivation and (**g**) V$_{50}$ of slow inactivation in mock- and TMEM233-transfected HEK293-Na$_V$1.7 cells. **h** Time-dependence and (**i**) time constant τ$_1$ of recovery from fast inactivation in mock- and TMEM233-transfected HEK293-Na$_V$1.7 cells. **j** Peak currents ($I_{max}$, nA) and (**k**) representative traces of currents elicited by slow ramp depolarizations (−100 mV to +20 mV at 1 mV/s) in control and TMEM233-

transfected HEK293-Na$_V$1.7 cells. **l** Peak currents of the last depolarization (30th pulse), normalized to the first pulse, from control and TMEM233-transfected HEK293-Na$_V$1.7 cells at pulse frequencies of 1, 2, 5, 10, and 20 Hz. **m** Left: Time constant τ$_1$ of recovery from fast inactivation in HEK293-Na$_V$1.7 cells that were mock-transfected (control), transfected with full-length TMEM233 (TMEM233) or N-terminally truncated TMEM233 lacking residues 1–34 (del. 1–34). Right: Time constant τ$_1$ of recovery from fast inactivation in HEK293-Na$_V$1.7 cells with buffer control (0.1% BSA) or synthetic TMEM233 N-terminal peptide (100 μM) in the intracellular solution (ICS). Data are shown as mean ± SEM; n.s., not significant; *$p < 0.05$. $n$ values and statistical information are detailed in Supplementary Table 1. Source data are included as a Source Data file.

tertiary structure of this peptide family, stabilized by three intramolecular disulfide bonds in an inhibitory cysteine knot motif, makes the gympietides plant-derived "knottin" Na$_V$ modulators and highlights a remarkable case of convergent evolution between algesic animal and plant venoms.

However, perhaps the most intriguing characteristic of the gympietides is their mechanism of action, most notably our observation that ExTxA has virtually no effect on Na$_V$1.7 function in heterologous expression systems and in neurons that do not express TMEM233[21], a transmembrane protein with hitherto undefined function. Although

the precise toxin binding site on TMEM233 remains to be determined, it is likely that the hydrophobic ExTxA, which we show binds to model membranes, interacts with extracellular or intramembrane residues that may include hydrophobic amino acids in the predicted TMEM233 re-entrant loop. A binding pocket buried in the membrane may also explain the near-irreversible effects of ExTxA on Na$_V$ function, and in turn the extremely long-lasting biological effects of *Dendrocnide* stings that include allodynia and painful flares that can, for example, be triggered by scratching of the sting site for weeks or even months following envenomation[8,9]. It is likely that the TMEM233-ExTxA

interaction serves to potentiate an otherwise low affinity toxin effect on $Na_V1.7$, albeit the molecular basis of this interaction remains to be determined. Thus, TMEM233 appears to act as a bifunctional molecule that binds ExTxA via the extracellular C-terminus or the transmembrane residues, and $Na_V1.7$ via at least the cytoplasmic module including the N-terminus, to produce an allosteric effect on gating of the channel upon binding of the peptide toxin.

Sequence alignment studies and phylogenetic analysis have characterized TMEM233 as a member of the dispanins, specifically subfamily B, with the two most closely related members PRRT2 and TRARG1 (formerly known as TUSC5) being notable for their substantially larger N-terminal domains. The limited sequence similarities of the dispanin subfamily B in this region, together with our observation that ExTxA also modulates $Na_V1.7$ function when co-expressed with PRRT2 or TRARG1, lends further support to the hypothesis that ExTxA interacts with extracellular or intramembrane residues in the more conserved C-terminal part of these proteins. While the precise motifs involved in the tripartite toxin-TMEM233-$Na_V1.7$ interaction remain to be determined, the profound ExTxA effects on inactivation parameters in particular suggest that domain IV of $Na_V1.7$ is likely involved in this process[22].

TMEM233 is notable for its expression in nociceptors, motivating the generation of a Tmem233[Cre] mouse in which Cre expression is under the control of *Tmem233* regulatory elements, with homozygotes being knockouts of the *Tmem233* gene[14]. Accordingly, toxin-induced pain behaviors were significantly reduced in Tmem233[Cre/Cre] knockout animals, with residual nocifensive responses likely explained by expression of, and activity of ExTxA at, Prrt2 and Trarg1 in overlapping neuronal populations[18]. Indeed, TRARG1 is expressed at high levels in human and mouse DRG, as well as in zebrafish sensory neurons[23,24]. While links to human disease, in particular painful conditions, have not yet been established for TMEM233, interestingly, several loss-of-function mutations of PRRT2 have been linked to multiple neurological diseases, including paroxysmal dyskinesias, benign familial infantile epilepsy, and hemiplegic migraine[25–27]. PRRT2 is known to interact with various synaptic vesicle proteins involved in neurotransmitter release, as well as the voltage-gated $Ca^{2+}$ subtype $Ca_V2.1$, which may contribute to human disease phenotypes associated with PRRT2 mutations[28]. Consistent with only minor or no effects of TMEM233 co-expression on $Na_V1.7$ function, nociceptive mechanical, heat and cold responses were unchanged in the Tmem233[Cre/Cre] knockout animals[14], albeit additional contributions to pathological pain states cannot be ruled out currently.

Our data suggest that in heterologous systems, TMEM233 can associate with $Na_V1.7$. Due to a lack of well-performing antibodies, it is currently not possible to directly confirm this interaction in native neurons, albeit it is strongly implied by ExTxA-induced effects on $Na_V1.7$ in DRG neurons and iPSC-derived human sensory neurons. Indeed, TRARG1 was previously identified as a $Na_V1.7$-interacting protein, likely due to the ubiquitous expression of TRARG1 in sensory neurons[18,23,29]. These observations suggest that TMEM233 is likely to be part of a native sensory neuron $Na_V$ signaling complex, based on our observations that it does not function as an ion channel itself, but that co-expression with $Na_V1.7$ confers sensitivity to algesic plant-derived knottin peptides.

$Na_V$ channels are known to function as multi-subunit complexes in native neurons, where they associate not only with auxiliary β subunits, but also several other proteins that may affect post-translational modifications, channel expression, trafficking, and function[30]. The β subunits are known to modulate interaction of various venom-derived peptides with the poreforming $Na_V$ α subunit; for example, the kinetics of inhibition by μ-conotoxins is affected by the presence of β subunits, while coexpression of β2 or β4 prevents inhibition by the μO§-conotoxin GVIIJ[31,32]. However, all previously studied $Na_V$-targeting toxins

appear to bind to the α subunit, making the gympietides toxins that require an interacting protein to modulate $Na_V$ channel function. In addition, the gympietides are also dispanin ligands, raising the possibility of future development of selective ligands that may find use as tool compounds to interrogate the function of this protein family, or that may ultimately lead to therapeutic applications. Finally, our study reports an association with $Na_V1.7$—a major player in peripheral pain signaling—and a function of TMEM233, providing a compelling impetus for further study of this intriguing transmembrane protein and its paralogs.

## Methods

### Peptide synthesis and purification

ExTxA and Pn3a peptide precursors were assembled via fluorenylmethyloxycarbonyl (Fmoc) solid-phase peptide synthesis on an automated microwave Liberty Prime synthesizer (CEM) using preloaded Fmoc-Wang (LL) resin (CEM, 0.31 mmol/g, 0.1 mmol scale). Amino acid couplings were performed in dimethylformamide (DMF) using 5 equivalents (eq) of Fmoc protected amino acid/5 eq Oxyma/10 eq N,N'-diisopropylcarbodiimide, relative to resin substitution, for 1 min at 105 °C. Removal of the Fmoc protecting group was achieved using pyrrolidine (25% v/v in DMF, 40 s at 100 °C). Cleavage and simultaneous removal of sidechain protecting groups were accomplished by treatment with 92.5% trifluoroacetic acid (TFA)/ 2.5% triisopropyl silane/ 2.5% $H_2O$/ 5% 2,2'-(ethylenedioxy)diethanethiol for 2 h at room temperature. TFA was evaporated by $N_2$ flow, following by peptide precipitation in cold diethyl ether and lyophilization in 0.1% TFA/ 50% acetonitrile (ACN)/ $H_2O$.

**Oxidative folding.** Linear ExTxA and analogues were dissolved in a minimum volume of aqueous ACN, then diluted to a peptide concentration of 0.25 mg/mL with 0.1 M $NH_4HCO_3$/ 50% iPro buffer (pH 8) containing 100 eq GSH/ 10 eq GSSG and stirred for 24 h at room temperature. The reaction was stopped by adding 1% TFA/ $H_2O$ to pH -5, and the correctly folded peptide was isolated by high-performance liquid chromatography (HPLC). Pn3a was oxidized at 0.1 mg/mL in 4.5 M $NH_4OAc$ at pH 8 with 2 M urea, and 1:10 oxidized to reduced glutathione (48 h at 4 °C). Oxidation solutions were acidified to pH <4 with TFA, filtered, and purified by RP-HPLC. Purity and correct mass were confirmed by RP-HPLC and ESI-MS in conjunction with 1D and 2D $^1H$ nuclear magnetic resonance (NMR), respectively.

**Click chemistry.** Folded [K(N3)]ExTxA (450 μM) and fluor 488 alkyne (BroadPharm, 670 μM) were dissolved in 50 mM potassium phosphate/ 30% DMSO (pH 7) containing 500 μM $CuSO_4$, 500 μM trishydroxypropyltriazolylmethylamine, 5 mM aminoguanidine and 5 mM sodium ascorbate and allowed to react at 37 °C for 1 h. The reaction was stopped by TFA 1% v/v and purified by HPLC.

**Biotinylation.** Folded ExTxA (870 μM) and biotin-PEG4-NHS (BroadPharm, 8700 μM) were first dissolved in minimal DMSO then diluted with $NaHCO_3$ (100 mM, pH 7) and allowed to react at 37 °C for 1 h. The reaction was stopped by TFA 1% v/v and purified by HPLC.

**Reversed-phase high-performance liquid chromatography (RP-HPLC).** Preparative and analytical HPLC were carried out on a Shimadzu LC-20AT system equipped with SPD-20A Prominence UV/VIS detector, an SIL-20AHT auto-injector. An Eclipse XDB – C18 column (Agilent; 5 μm, 9.4 cm × 250 mm, 300 Å, flow rate 5 mL/min) with gradient from 45–75% solvent B in 60 min was used to purify the oxidized peptides. A Zorbax 300SB – C18 column (Agilent; 5 μm, 4.6 cm × 250 mm, 300 Å, flow rate 1 mL/min) with gradient from 45–75% solvent B in 60 min was used to purify tagged peptides. A Hypersil GOLD – C18 column (ThermoFisher; 3 μm, 2.1 × 100 mm, 175 Å, flow rate 0.7 mL/min) with gradient 0–80%B over 16 min was used to monitor

oxidation and ligation, and to analyze the purity of the peptides. Absorbance was monitored at 214 nm and 280 nm. Solvent A: 0.05% TFA in $H_2O$; solvent B: 90% ACN/ 0.043% TFA in $H_2O$.

**Mass spectrometry (MS).** A Q-Star Elite mass spectrometer (Applied Biosystems, MDS SCIEX) equipped with Shimadzu LC-20AB pump and an SIL-20ACHT auto-injector was used for mass analysis. Samples (50 μg) were dissolved in 50% aqueous ACN (100 μL), then 5 μL of the samples were injected into the spectrometer for analysis by Analyst QS Software 2.0. A Zorbax 300SB – C18 column (Agilent; 3.5 μm, 2.1 × 100 mm, 300 Å, flow rate 0.25 mL/min) was used to run all samples. The TOF mass range was set at 500–1800 Da. The gradient program was as follow: 0–0.5 min, 2% solvent B; 0.5–10 min, 2-60% solvent B; 10–11 min, 60–96% solvent B; 11–13 min, at 96% solvent B; and 13.10–20 min, 2% solvent B. Solvent A is 0.1% formic acid in H2O and solvent B is aqueous ACN with 0.1% formic acid.

**Recombinant expression of $^{15}$N-labelled ExTxA and nuclear magnetic resonance spectroscopy (NMR).** SHuffle® T7 Competent E. coli cells (New England Biolabs, Massachusetts, USA) were transformed with an expression vector encoding a 6xHis/SUMO solubility tag and ExTxA. The transformed cells were grown in Lysogeny Broth (LB) supplemented with ampicillin at 30 °C until OD600 = 0.8–1.2. The cells were then transferred to 15N-labeled minimal media M9 containing 90.2 mM Na2HPO4, 22.0 mM KH2PO4, 17.1 mM NaCl, 2 mM MgSO4, 0.1 mM CaCl2, 1x BME vitamin solution (Sigma-Aldrich, Missouri, USA), 1% (w/v) glucose, 18.7 mM 15NH4Cl and 100 μg/mL ampicillin before induction with 1 mM isopropyl β-D-1-thiogalactopyranoside (IPTG) and incubated at 30 °C for 3 h. Cells were pelleted at 5000 × *g* for 10 min and lysed by ultrasonication in buffer containing 40 mM Tris pH 8, 400 mM NaCl and 0.1 mg/mL lysozyme. Cell debris was removed by centrifugation at 40,000 × *g* for 30 min at 4 °C. Recombinant His6-SUMO-ExTxA fusion was isolated using a Ni-NTA resins (Cytiva, Massachusetts, USA) and was cleaved by addition of 400 μg/mL SUMO protease and 2 mM dithiothreitol (DTT). The cleaved peptide was then purified by RP-HPLC on a semipreparative C3 Agilent Zorbax column (Agilent, California, USA). The peptide was then lyophilized, dissolved in 6 M guanidine HCl and diluted 20 times with folding buffer containing 200 mM ammonium bicarbonate pH 8.2, 55% isopropanol, 5 mM GSH and 0.5 mM GSSG and left stirring overnight at room temperature. Refolded peptide was then purified by RP-HPLC using an analytical C18 Phenomenex Jupiter column (Phenomenex, California, USA).

For NMR analysis, lyophilized $^{15}$N-labelled ExTxA was resuspended either in 40% deuterated acetonitrile or buffer containing 20 mM Tris pH 8, 50 mM NaCl, 10% (w/v) dodecylphosphocholine (DPC) and 5% $D_2O$. $^1$H-$^{15}$N transverse relaxation optimized spectroscopy (TROSY) spectra were acquired for the peptide samples (30 μM) on a Bruker Avance 900 MHz NMR spectrometer equipped with a triple resonance cryogenic probe at 298 K. The spectra were processed using TOPSPIN v4.1.4 software (Bruker, Massachusetts, USA).

**Animals and ethics approvals**
All experiments involving animals were reviewed and approved by local Institutional animal ethics committees (US Veterans Affairs West Haven Medical Center Animal Care and Use Committee; University of Queensland Molecular Bioscience Animal Ethics Committee; King's College London Animal Welfare and Ethical Review Body; University College London Animal Welfare and Ethical Review Body) and conducted in accordance with relevant national and international regulations (International Association for the Study of Pain Guidelines for the Use of Animals in Research; the Australian Code of Practice for the Care and Use of Animals for Scientific Purposes, 8th edition (2013); UK Home Office Project Licence). C57BL/6 mice for in vivo experiments or tissue collection were sourced either from the Animal Resource Centre

(Canning Vale, Western Australia, Australia) or local breeding colonies. Animals of both sexes were used. Experiments were not designed or powered to detect sex-specific differences. Animals were housed in groups of three or four per cage under 12-h light-dark cycles and had standard rodent chow and water *ad libitum*.

Tmem233$^{Cre}$ animals were generated and genotyped as previously described, with a Kozak-Cre-pA cassette replacing the ATG start codon of the murine Tmem233 gene to produce a functional knockout in homozygous animals[14]. Na$_V$1.7$^{Advill}$ mice lacking functional Na$_V$1.7 in all sensory neurons were generated and genotyped as previously described[33] by crossing Advillin-Cre mice with floxed Scn9a animals in which loxP sites were inserted in introns flanking exons 14 and 15[34,35]. Na$_V$1.9$^{-/-}$ mice and Na$_V$1.8$^{Cre/Cre}$ mice, in which Cre recombinase substitutes the translational start-site of Na$_V$1.8, were generated and sequenced as previously described[34,36,37].

*X. laevis* were purchased from ENASCO, Fort Atkinson, WI 53538-0901, USA. *X. laevis* oocyte surgeries were reviewed and approved by the Anatomical Biosciences group of the Animal Ethics Committee at The University of Queensland (QBI/AIBN/087/16/NHMRC/ARC) and conducted in accordance with Australian quarantine regulations.

**Dorsal root ganglion (DRG) neuron isolation and culture**
DRG from all spinal levels were isolated from male and female mice (wild-type C57BL/6 J, Tmem233$^{Cre/Cre}$, Na$_V$1.8$^{Cre/Cre}$, Na$_V$1.9$^{-/-}$ or littermate controls) as previously described[36,38,39]. For $Ca^{2+}$ imaging experiments, DRG neurons were dissociated following incubation with 1% collagenase type IV (Worthington, New Jersey, USA) for 1.5 h (37 °C, 5% $CO_2$) and 2.5% trypsin (Sigma-Aldrich, Montana, USA) for 15 min and resuspended in Dulbecco's modified Eagle's medium (DMEM) (Sigma-Aldrich, New South Wales, Australia) supplemented with 10% fetal bovine serum (FBS) (Thermofisher, Massachusetts, USA) and penicillin/streptomycin (Thermofisher, Massachusetts, USA). Following plating in 96-well poly-D-lysine–coated culture plates, isolated neurons were incubated for 24 h at 37 °C with 5% $CO_2$ prior to imaging.

For manual patch clamp experiments in DRG neurons from Tmem233$^{Cre/Cre}$ mice, DRGs isolated from male and female homozygous Tmem233$^{Cre/Cre}$ mice and littermate controls were placed in 2.5% collagenase type IV (Worthington, New Jersey, USA) for 3 h (37 °C, 5% $CO_2$). Whole DRG were then exposed to 0.25% Trypsin (Sigma-Aldrich) for 15 min prior to trituration with a fire-polished glass Pasteur pipette. The cell suspension was triturated and observed under a microscope until whole ganglia were reduced to single cells. The single cell suspension was then treated briefly with 0.05% DNase (Worthington) and passed through (1000 rpm) a cushion of bovine serum albumin (15% w/v) in Minimum Eagle Media (MEM) to remove debris. Poly-D-lysine (PDL) and laminin pre-coated borosilicate glass coverslips (Corning, NY, USA) were used to plate out isolated single-cell DRG neurons which were left to adhere for >90 min. Finally, coverslips were flooded with MEM 10% FBS, 10 μM cytosine arabinoside, 1% v/v penicillin and streptomycin, 50 ng/mL NGF (Promega, Southampton, UK) and 50 ng/mL GDNF (Promega). DRG cultures were used the following day (~16−30 h post isolation) for whole-cell recordings.

For manual patch clamp experiments in DRG neurons from Na$_V$1.8$^{Cre/Cre}$ and Na$_V$1.9$^{-/-}$ mice, DRGs were incubated in 0.5 U/mL Liberase TM (Sigma-Aldrich, St. Louis, MO) and 0.6 mM EDTA for 20 min at 37 °C, followed by a 15-min incubation at 37 °C in 0.5 U/mL Liberase TL (Sigma-Aldrich), 0.6 mM EDTA, and 30 U/mL papain (Worthington Biochemical, Lakewood NJ). Following collection by centrifugation, DRGs were triturated in Dulbecco's modified Eagle's medium/F12 (1:1) with 100 U/mL penicillin, 0.1 mg/mL streptomycin (Life Technologies, Grand Island, NY), and 10% fetal bovine serum (Hyclone, Logan, UT), containing 1.5 mg/mL bovine serum albumin (low endotoxin; Sigma-Aldrich) and 1.5 mg/mL trypsin inhibitor (Sigma-Aldrich). The single cell suspension was then plated in DRG media containing 1.5 mg/mL bovine serum albumin (low endotoxin; Sigma-Aldrich) and 1.5 mg/mL

trypsin inhibitor (Sigma-Aldrich) on poly-d-lysine/laminin-coated coverslips (Corning, Discovery Labware, Bedford, MA).

## Cell lines and culture

Cell lines were obtained from commercial suppliers and not authenticated.

**TE-671 cells.** TE-671 cells (CRL-8805; European Collection of Cell Cultures, Porton Down, Salisbury, United Kingdom) were cultured in MEM supplemented with 10% (v/v) FBS and 2 mM L-glutamine (Thermofisher, Massachusetts, USA).

**HEK293 cells.** Human Embryonic Kidney (HEK) 293 cells (85120602; American Tissue Culture Collection, Manassas, VA, USA) used for transient transfections were maintained in high-glucose Dulbecco's Modified Eagle Medium (DMEM) supplemented with 10% fetal bovine serum (FBS), l-glutamine (2 mM), pyridoxine and sodium pyruvate (110 mg/mL). Cells were split every 4–6 days using TrypLE Express (Invitrogen, Massachusetts, USA) and cultured in a 37 °C/5% $CO_2$ incubator. For transient transfection of $Na_V1.7$ constructs, HEK293 cells stably expressing the β1 and β2 subunit under G418-selection were used[40].

*Stable HEK293-$Na_V1.1$, HEK293-$Na_V1.2$, HEK293-$Na_V1.3$, HEK293-$Na_V1.4$, HEK293-$Na_V1.5$, HEK293-$Na_V1.6$, HEK293-$Na_V1.7$, CHO-$Na_V1.7$ and CHO-$Na_V1.8$ cells.*

HEK293 cells stably expressing $hNa_V1.1$/β1 – $hNa_V1.7$/β1 (SB Drug Discovery, Glasgow, United Kingdom) were maintained in MEM supplemented with 10% (v/v) FBS, L-glutamine (2 mM), and selection antibiotics (geneticin (600 µg/mL, Sigma-Aldrich, Missouri, USA) and blasticidin (4 µg/mL, Thermofisher, Massachusetts, USA) for all except HEK293-$hNa_V1.4$ cells which were grown in medium additionally containing zeocin (500 µg/mL, Thermofisher, Massachusetts, USA)). Chinese Hamster Ovary (CHO) cells stably expressing $hNa_V1.7$ in the absence of β subunits (ChanTest, Ohio, USA) and CHO cells stably expressing tetracycline-inducible $hNa_V1.8$/β3 (ChanTest, Ohio, USA) under hygromycin and zeocin selection were cultured in MEM additionally supplemented with 10% (v/v) FBS and 2 mM L-glutamine. Tetracycline (1 µg/mL, Sigma-Aldrich, Missouri, USA), freshly prepared from frozen stocks, was added a minimum of 48 h prior to functional assays to induce $hNa_V1.7$ and $hNa_V1.8$ expression. All cell lines were grown in a humidified 5% $CO_2$ incubator at 37 °C and passaged using TrypLE Express.

**ND7/23 cells.** The ND7/23 DRG neuroblastoma hybridoma line (92090903) was obtained from Sigma–Aldrich (Castle Hill, New South Wales, Australia) and maintained in high glucose DMEM containing 10% heat-inactivated FBS, l-glutamine (2 mM), pyridoxine and sodium pyruvate (110 mg/mL). Confluent cultures were dissociated using TrypLE Express and grown in a humidified 5% $CO_2$ incubator at 37 °C.

**F11 cells.** F11 DRG x neuroblastoma hybridoma cells (08062601; Sigma Aldrich Australia, European Cell Culture Collection) were cultured at 37 °C/5% $CO_2$ in Ham's F12 media containing 10% FBS, 100 µM hypoxanthine, 0.4 µM aminopterin and 16 µM thymidine (HAT media supplement Hybri-Max™, Sigma–Aldrich, Castle Hill, Australia) and passaged using TrypLE Express.

**SH-SY5Y cells.** SH-SY5Y human neuroblastoma cells (94030304; Sigma Aldrich Australia, European Cell Culture Collection) were maintained at 37 °C/5% $CO_2$ in Roswell Park Memorial Institute (RPMI) medium containing L-glutamine (2 mM) and 15% FBS, and passaged every 5–7 days using TrypLE Express.

**Human iPSC-derived sensory neurons.** Human iPSC-derived sensory neurons (RealDRG™) produced by Anatomic Inc. from the female subject ANAT001 were differentiated to produce immature sensory neurons with scaled-up versions of Anatomic's Senso-DM kit. Cryopreserved neurons at day 7 post-differentiation were obtained from Anatomic Inc., plated on glass coverslips coated with poly-L-ornithine (Sigma, Castle Hill Australia) and Matrix 3 (Anatomic Inc., Minneapolis, USA) in 12-well tissue culture plates (Corning) at a density of 50,000 cells/well and maintained in Chrono™ Senso-MM maturation medium (Anatomic cat# 7008) for another 7–21 days (DIV14–28). Growth media was exchanged three times a week and recordings performed between DIV14–28.

## Plasmids and transfection

Plasmids encoding the human $Na_V$ β1,2,3 and 4 subunits (SCN1B transcript variant a; SCN2B; SCN3B transcript variant 1 and SCN4B transcript variant 1) were purchased from OriGene (Rockville, MD, USA) and subcloned into a pIRES-EGFP-Puro plasmid (Addgene, Massachusetts, USA) by the University of Queensland Protein Expression facility, resulting in a construct leading to separate expression of the β subunits and an EGFP/puromycin resistance fusion protein following the IRES2 element. Custom pSpCas9(BB)-2A-Puro PX459 plasmids encoding three separate guide RNAs which target TMEM233 (5′-TGCGTACCCCATCAACATCG-3′, 5′-GTCTCAGTACGCCCCTAGCC-3′ and 5′-TCTCTGAACAGCTACAACGA-3′) were purchased from GenScript (Nanjing, Jiangsu Province, China). pcDNA3.1 plasmids encoding hTMEM233, N-terminal HA-tagged TMEM233, C-terminal HA-tagged TMEM233, del.1-34 TMEM233, FLAG-tagged $hNa_V1.7$, hPRRT2, hTRARG1, GPAA1, PIGK, PIGS, PIGT, PIGU, CRELD1, LMAN2L, MMGT1, RNF121, and STT3B were purchased from GenScript (Singapore) or obtained through the FANTOM Riken cDNA initiative. Plasmids were prepared using Qiagen HiSpeed Maxi Prep or Qiagen Endofree Maxi Prep kits (Qiagen, Hilden, Germany) as per the manufacturer's protocol. For transfection, cells were seeded at 70–80% confluency in T25 or T75 flasks and transfected with Lipofectamine 2000 or Lipofectamine 3000 according to the manufacturer's protocols. Empty pcDNA3.1 plasmid was used as a mock transfection control or to adjust total DNA amount to 1.8 µg (T25) or 5.4 µg (T75). Cells were cultured for 24–48 h in normal growth medium prior to functional assays, except for CHO cells transfected with β1-β4 plasmids, and TE-671 cells transfected with TMEM233 guide RNAs/Cas9, which were cultured under puromycin selection for approximately 4 weeks prior to assays. Stable HEK293-$Na_V1.7$/C-terminal HA taggedTMEM233 cells were generated by clonal selection using hygromycin (50 µg/mL in complete growth medium) following transfection with a pcDNA3.1/Hygro(+)-TMEM233 C-HA plasmid.

## Electrophysiology

**Automated patch-clamp electrophysiology.** Whole-cell patch clamp experiments involving HEK293, COS-1, CHO, ND7/23, F11, SH-SY5Y and TE-671 cells were performed on a QPatch II automated electrophysiology platform (Sophion Bioscience, Ballerup, Denmark) as previously described[38]. Cells at approximately 70–80% confluency were harvested from T75 flasks using TrypLE Express at 37 °C for 2–5 min and resuspended in DMEM with 25 mM HEPES (Sigma-Aldrich, NSW, Australia), 100 U/mL Penicillin-Streptomycin (Gibco) and 40 µg/mL trypsin inhibitor from Glycine max (soybean) (Sigma Aldrich) prior to recovery on a QStirrer (Sophion) for 30 min. 16-channel patch plates with a patch hole diameter of 1 µm and resistance of 2 ± 0.02 MΩ were used. Cell positioning and sealing parameters were as follows: positioning pressure −60 mbar, minimum seal resistance 0.1 GΩ, holding potential −100 mV, holding pressure −20 mbar. Whole-cell currents were filtered at 5 kHz and acquired at 25 kHz. Series resistance compensation was used and set to 70%.

The composition of the extracellular solution (ECS) was (in mM): 145 NaCl, 4 KCl, 2 $CaCl_2$, 1 $MgCl_2$, 10 HEPES, and 10 glucose (pH 7.4) (osmolarity, 305 mOsm). For experiments in HEK293 cells expressing

Na$_V$1.4, Na$_V$1.5 and Na$_V$1.7, the concentration of NaCl was reduced to 70 mM and replaced with 75 mM choline chloride. The composition of the intracellular solution was (in mM): 140 CsF, 1/5 EGTA/CsOH, 10 HEPES, and 10 NaCl (pH 7.3) with CsOH (osmolarity, 320 mOsm).

**IV curve and voltage-dependence of steady-state fast inactivation protocols.** Current-voltage (IV) curves were determined in HEK-Na$_V$1.7 cells, in the absence and presence of TMEM233, from a series of step pulses (500 ms) ranging from −110 to +55 mV (5 mV increments, repetition interval 5 s) before and after incubation with ExTxA (1 μM) from a holding potential of −90 mV. Currents were converted to conductance (G) using G = I/(V − V$_{rev}$), with V$_{rev}$ being the reversal potential, plotted against voltage and fitted with a Boltzmann equation to obtain GV (conductance-voltage) curves. Voltage dependence of steady-state fast inactivation was determined by a subsequent 10 ms pulse to −20 mV immediately following the 500 ms depolarization step detailed above to assess the available non-inactivated channels.

**ExTxA pharmacology.** Effects of ExTxA on Na$_V$ current in heterologous expression systems or neuronal cell lines were assessed following incubation with varying concentrations of ExTxA (0.1 nM – 1 μM), Pn3a (100 nM) and TTX (1 μM) diluted in ECS with 0.1% BSA for 5 min at a holding potential of −90 mV. Na$_V$ current was elicited by depolarization (50 ms) to −20 mV every 20 s (0.05 Hz), with persistent current (40 ms from peak current) normalized to peak current as specified.

**Voltage-dependence of slow inactivation.** To determine the voltage dependence of slow inactivation, HEK293-Na$_V$1.7 cells were held at −120 mV and peak current determined by an initial 20 ms pulse to 0 mV. Following a series of 15,000 ms step pulses ranging from −120 to 0 mV in 20 mV increments, a subsequent 50 ms test pulse to 0 mV, preceded by a 50 ms step to −120 mV to remove fast inactivation, was used to determine the remaining current. The proportion of slow-inactivated channels was determined by normalizing peak current from the test pulse to the initial depolarization. A Boltzmann equation was fit to the current data, plotted as a function of voltage, to determine the V$_{50}$ as well as the minimum proportion of activatable channels.

**Ramp current protocols.** The holding potential was −90 mV. After a 50 ms pulse to −100 mV, ramps from −100 mV to +20 mV at a rate of 0.2 and 1.0 mV/ms were applied before and after ExTxA (100 nM) addition.

**Use-dependence protocol.** Use-dependence experiments were performed on the Patchliner Quattro (Nanion Technologies GmbH, Munich, Germany) using an EPC 10 USB Quadro Patch Clamp Amplifier (HEKA Elektronik GmbH, Lambrecht/Pfalz, Germany) and PatchControlHT 2.01.30 and PatchMaster v2x90.4 beta softwares (HEKA Elektronik GmbH, Lambrecht/Pfalz, Germany). Cells at approximately 70–80% confluency were harvested from T75 flasks using Accutase (Thermofisher, Massachusetts, USA) at room temperature for 2–5 min and resuspended in high-glucose Dulbecco's Modified Eagle Medium (DMEM) (Sigma-Aldrich, New South Wales, Australia) with 15 mM HEPES (Sigma-Aldrich, NSW, Australia) prior to recovery in the cell hotel of the Patchliner for 10 min. ECS contained (in mM): NaCl 140, KCl 4, CaCl$_2$ 2, MgCl$_2$ 1, HEPES 10, glucose 5. The pH was adjusted to 7.4 with NaOH, and the osmolarity was 298 mOsm. ICS contained (in mM): CsCl 50, NaCl 10, CsF 60, EGTA 20, HEPES 10. The pH was adjusted to 7.2 with CsOH, and the osmolarity was 285 mOsm. A seal enhancer was used containing (in mM): NaCl 80, KCl 3, MgCl$_2$ 10, CaCl$_2$ 35, HEPES Na$^+$ salt 10. The pH was adjusted to 7.4 with HCl, and the osmolarity was 298 mOsm. The seal enhancer was washed off before starting the experiments. HEK293 cells stably expressing hNa$_V$1.7/ β1 or hNa$_V$1.7/ β1/C-HA TMEM233 were used. Consecutive series of 30 depolarizing

20 ms pulses from −90 mV to −20 mV were applied with frequencies of 1 Hz, 2 Hz, 5 Hz, 10 Hz, and 20 Hz. After the experiment ExTxA (100 nM) was applied to confirm TMEM233 expression. The amplitude of the test current was normalized to the current induced by the first pulse of its series, with the last test pulse plotted as a function of frequency for statistical analysis.

**Recovery from fast inactivation protocol.** To determine recovery from fast inactivation, a 20 ms pulse to 0 mV from a holding potential of −90 mV was used to induce fast inactivation. A subsequent recovery period, ranging from 1 ms to 2216 ms in 50% increments at −90 mV, was followed by a last 20 ms test pulses to 0 mV to determine the time-dependence of recovery from fast inactivation. The proportion of available channels was determined by normalizing peak currents from the last to the first pulse. A two-phase exponential equation was fit to the normalized current, plotted as a function of time, to determine the time constants τ$_1$ and τ$_2$ of recovery from fast inactivation.

### Xenopus laevis oocyte electrophysiology

Human Na$_V$1.7 and TMEM233 genes cloned into pcDNA3.1 plasmid were linearized with NotI restriction enzyme and used as template for cRNA synthesis using T7 polymerase (mMessage mMachine Kit; Life Technologies). Stage V/VI oocytes were injected with 5–20 ng total RNA, then incubated at 17 °C for 1–3 days in 50% Leibovitz's L-15 medium (Gibco), supplemented with 25 μg/mL gentamicin, 25 μg/mL streptomycin, and 2.5% fetal horse serum. Two-electrode voltage clamp (Axoclamp 900 A amplifier; Axon Instruments) were performed at room temperature in ND96 solution (in mM: 96 NaCl, 2 KCl, 1 MgCl2, 1.8 CaCl2, 5 HEPES; pH 7.4 with NaOH) with a ~40 μL recording chamber. In experiments where ExTxA was added, solutions were supplemented with 0.05% fatty-acid free bovine serum albumin. Borosilicate glass microelectrodes had resistances of 0.2–0.8 MΩ when backfilled with 3 M KCl. Data were digitized at 20 kHz and filtered at 2 kHz using pCLAMP 11 software (Digidata 1550B; Axon Instruments).

ExTxA activity was assayed at 1 μM, analyzing the current from a 50 ms step to −10 mV every 10 s, from a holding volage of −80 mV. For all other voltage protocols, oocytes were held at −90 mV. Activation curves were obtained by measuring peak currents during steps from −100 mV to +60 mV in +10 mV increments applied every 10 s, with G/G$_{max}$ plot between −80 mV and +10 mV. Steady-state inactivation curves plot the peak current at a test pulse of −10 mV, after a 200 ms pre-pulse from −100 mV to +40 mV in +10 mV increments applied every 10 s. Slow inactivation curves plot the peak current at a test pulse of −10 mV, after a 10 s pre-pulse from −110 mV to −10 mV in +10 mV increments applied every 20 s. To determine recovery from inactivation, oocytes were first stepped to −10 mV, then rested at −90 mV for a variable time interval indicated in the graph, then a second −10 mV depolarization applied to determine the fraction of channls recovered. This protocol was applied every 13 s (time between each initial −10 mV step).

**Manual patch-clamp electrophysiology in DRG neurons.** To determine the effects of ExTxA on total and TTX-s Na$_V$ current in wild-type DRG neurons, whole cell voltage-clamp recordings were made 20–48 h following DRG isolation using a Multiclamp 700B amplifier (Molecular Devices) and Clampex 10.7 software suite, with data sampled at 10 kHz and filtered at a frequency of 2 kHz. Pipette resistance was kept between 1–3 MOhm. Internal solutions contained (in mM): CsF 140, NaCl 10, EGTA 1, HEPES 10, D-glucose 10 (pH 7.3 with CsOH, osmolarity 310–315). External solutions contained (in mM): NaCl 50, Choline-Cl 90, KCl 3, HEPES 10, MgCl$_2$ 1, CaCl$_2$ 1, TEA-Cl 20, CdCl$_2$ 0.1, CsCl 5 (pH 7.4 with NaOH, osmolarity 320–330). Currents were evoked by consecutive depolarization steps to −20 mV (50 ms) from a holding potential of −90 mV every 10 s in the presence of buffer control (0.1% BSA) or ExTxA (100 nM in 0.1% BSA) followed by TTX (1 μM in 0.1% BSA)

or ExTxA + TTX. Persistent currents at 40 ms were normalized to peak current for analysis.

The effect of ExTxA on $Na_V1.8$ channels was examined in $Na_V1.9$-null dorsal root ganglion (DRG) neurons. Electrophysiological recordings were conducted at room temperature (~22 °C) 24 hr after DRG neuron isolation using an Axopatch 200B amplifier (Molecular Devices). Pipette potential was adjusted to zero before seal formation, and liquid junction potential was not corrected. Capacity transients were canceled and voltage errors were minimized with 80–90% series resistance compensation. Voltage-dependent currents were acquired with Clampex 10.7, sampled at 50 or 100 kHz, and filtered at 5 kHz. The extracellular solution (ECS) contained (in mM): 50 NaCl, 90 choline•Cl, 3 KCl, 1 $MgCl_2$, 1 $CaCl_2$, 10 HEPES, 5 CsCl, 20 TEA·Cl, 0.1 $CdCl_2$, pH 7.4 with NaOH (~326 mOsmol/L). At the start of the experiment, 4-aminopyridine (4 mM) and tetrodotoxin (TTX, 1 µM) were added to the bath solution to block endogenous potassium channels and TTX-sensitive sodium channels. The internal pipette solution contained (in mM): 140 CsF, 10 NaCl, 1 EGTA, 10 HEPES, and 10 dextrose, pH 7.3 with CsOH (adjusted to 311 mOsmol/L with sucrose). Small-diameter DRG neurons (24–30 µm) were selected because they produce large $Na_V1.8$ currents. Cells were held at −100 mV and $Na_V1.8$ currents were elicited with a series of depolarizations (0 mV for 100 ms, 10-sec intervals). Once the current amplitude became stable, ExTxA (100 nM in 0.1% BSA, 10 ml) or vehicle control (0.1% BSA, 10 mL) was applied by superfusion. The persistent currents were normalized to peak currents, and the effects on persistent current were evaluated by comparing the change of the persistent current at 40 ms (persistent current amplitude after ExTxA or BSA treatment minus persistent current amplitude before ExTxA or BSA treatment). Finally, to confirm ExTxA-sensitivity of recorded neurons, extracellular TTX was washed away by normal ECS at the end of experiment. Only cells with large persistent currents in response to TTX wash-off, likely produced by $Na_V1.7$ channels, were included for data analysis.

The effect of ExTxA on $Na_V1.9$ was examined in $Na_V1.8$-null ($Na_V1.8^{Cre/Cre}$) DRG neurons as detailed above for $Na_V1.8$ currents. The extracellular solution (ECS) contained (in mM): 140 NaCl, 3 KCl, 1 $MgCl_2$, 1 $CaCl_2$, 10 HEPES, 5 CsCl, 20 TEA·Cl, 0.1 $CdCl_2$, pH 7.4 with NaOH (327 mOsmol/L). The internal pipette solution contained (in mM): 140 CsF, 10 CsCl, 5 EGTA, 10 HEPES, 2 Na-adenosine 5′-triphosphate, 0.3 Na-guanosine 5′-triphosphate, 2.5 Na-creatine phosphate, and 9 dextrose, pH 7.3 with CsOH (311 mOsmol/L). Small-diameter DRG neurons (<30 µm) were selected, and $Na_V1.9$ currents were evoked by two 50 ms depolarizations of −50 and −40 mV after a 500 ms prepulse of −130/−140 mV before each depolarization (2-pulse protocol, 20-sec interval). There was a time-dependent increase in $Na_V1.9$ currents with maximal peak amplitude achieved around 10 min after forming the whole-cell configuration. Cells producing more than 1 nA $Na_V1.9$ currents with little/no run-down were superfused with ECS followed by 10 mL ECS containing either 0.1% BSA (control) or ExTxA (1 µM in 0.1% BSA). Data were analyzed using Clampfit 10.7 (Molecular Devices) and OriginPro 2021b (Microcal Software) and presented as mean ± SEM unless stated otherwise.

Whole-cell patch clamp experiments in $Tmem233^{Cre/Cre}$ DRGs and DRGs from wild-type littermates were obtained using an Axopatch 200B amplifier (Molecular Devices, CA, USA) and signals digitized using Digidata 1440 A (Molecular Devices), data were visualized and analyzed using the pClamp software suite (Version 10). Voltage-clamp internal solution contained (in mM); CsMethanesulfonate 130, NaCl 20, EGTA 0.2, HEPES 10, Mg-ATP 4, Na-GTP 0.3 and dextrose 10. The external solution contained (in mM): NaCl 30, Choline-Cl 110, KCl 3, $MgCl_2$ 1, $CaCl_2$, HEPES 10, CsCl 5, TEA-Cl 20, $CdCl_2$ 0.1 and 4-AP 1. External solution contained (in mM) NaCl 140, KCl 3, MgCl2 2, CaCl2 2, HEPES 10. Internal and external solutions were pH 7.2 adjusted with CsOH or NaOH and osmolarity was corrected using D-glucose, 300–310; 310–320 mOsm, respectively. Pipettes were pulled using

borosilicate glass (1.5 OD, 1.17 ID, Harvard Apparatus, UK) in a Narashige (PC-10, UK) pipette puller and fire polished (Mircoforge MF830, Narashige) on the day of experiment. Pipette resistance was kept between 1–3 MOhm and after whole-cell configuration was achieved neurons were held at −80 mV. Voltage errors were kept below 10% using 80–90% series resistance compensation. Recordings were acquired at 25 kHz with a 2.9 kHz low-pass Bessel filter and recordings included a P/6 linear leak subtraction. Prior to recording, coverslips were incubated with isolectin GS-IB4 from *Griffonia simplicifolia*-AlexaFluor488 conjugate (1 µg/µL) for 5 min then washed twice with external solution to visualized IB4 + and IB4- DRG neurons[41]. DRG on coverslips were placed in a low-volume (<500 µL) bath (Warner Instruments) for recordings. Following a control (before) recording, taken 5 min post break-in, ExTxA (30 nM, 0.1% BSA) in ECS was perfused for 2–3 min through a gravity-driven perfusion system to allow sufficient fluid exchange and toxin binding. Pulses to −20 for 100 ms were used to assess persistent current (average current amplitude during final 30 ms) following ExTxA exposure.

**Manual patch-clamp electrophysiology in human iPSC-derived sensory neurons.** Whole-cell patch clamp experiments in human iPSC-derived sensory neurons were conducted between DIV14-28 using Multiclamp 700B amplifier (Molecular Devices) and Clampex 10.7 software suite, with data sampled at 10 kHz, filter frequency of 2 kHz, and 50–60% series resistance compensation. Pipette resistance was kept between 1–3 MOhm. External solutions contained (in mM): NaCl 140, KCl 3, HEPES 10, $MgCl_2$ 1, $CaCl_2$ 1, TEA-Cl 20, $CdCl_2$ 0.1, CsCl 5 (pH 7.4 with NaOH, osmolarity 320–330). Internal solutions contained (in mM): CsF 140, NaCl 10, EGTA 1, HEPES 10, D-glucose 10 (pH 7.3 with CsOH, osmolarity 310–315). Cells were held at −100 mV and currents were evoked by consecutive depolarizations to −20 mV (50 ms, 10 s intervals). Vehicle control (ECS containing 0.1% BSA), ExTxA (100 nM), Pn3a (100 nM), and TTX (1 µM) were washed on consecutively. Persistent currents at 40 ms were normalized to peak current for analysis.

### TKOv3 screen in TE-671 cells

As previously described[42], approximately 70–100 million TE-671 cells were prepared across five T-175 flasks. Polybrene (8 µg/mL; Sigma-Aldrich, Castle Hill, Australia) was added to the growth medium and subsequently, Toronto KnockOut (TKO) CRISPR Library version 3 lentivirus encoding 70948 guides (4 gRNA/gene) (Addgene, Massachusetts, USA) was added to each flask (volume yielding a multiplicity of infection of approximately 0.5 was used). Cells were incubated for 24 h at 37 °C and 5% $CO_2$. Cell medium was replaced with fresh growth medium, and puromycin (10 µg/mL) added after 24–48 h for the selection of transduced cells. TE-671 cells transduced with TKO v3 lentivirus library were passaged 6 times under selection before separation into three groups, each containing approximately 70–100 million cells across multiple cell culture flasks. Cells were exposed to a combination of veratridine (5 µM) and ouabain (20 nM) with or without ExTxA (1 µM), or no selection for 3 days. Cells were washed with PBS, and fresh growth medium was added. Cells were allowed to recovery over a period of 3–5 days. Following recovery, cells underwent the selection process twice more. After the third round of selection, cells were harvested, and genomic DNA (gDNA) was extracted using ISOLATE II Genomic DNA Kit (Bioline, Meridian Bioscience, Ohio, USA).

**Genomic DNA sequencing.** Genomic DNA was extracted from harvested TE-671 cells and was used for PCR reactions. Primers used to amplify TKO v3 single guide RNAs (sgRNAs) were as follows; sense, 5′-GGA CAG CAG AGA TCC AGT TTG GT-3′ and antisense, 5′-GAG CCA ATT CCC ACT CCT TTC AA -3′. Amplification was carried out using NEBNext Q5U Master Mix (BioLabs New England, Massachusetts, USA) with 20 cycles (98 °C for 10 sec followed by 66 °C for 30 sec). PCR

products were purified using QIAquick PCR purification kit (QIAGEN, Hilden, Germany). Secondary PCR was performed to attach Illumina adaptors as well as to barcode samples. Primers for the secondary PCR include a 6 base pair barcode for multiplexing of varying biological samples and replicates as well as variable length sequences to increase library complexity: sense, 5′- AAT GAT ACG GCG ACC ACC GAG ATC TAC ACT CTT TCC CTA CAC GAC GCT CTT CCG ATC T (1–9 base pair variable length sequence) TCT TGT GGA AAG GAC GAA ACA CC-3′ and antisense; 5′- CAA GCA GAA GAC GGC ATA CGA GAT (6 base pair barcode) GTG ACT GGA GTT CAG ACG TGT GCT CTT CCG ATC TAC CGA CTC GTT GCC ACT TTT TCA AG-3′. Amplification was carried out using NEBNext Q5U Master Mix (BioLabs New England, Massachusetts, USA) with 3 cycles (98 °C for 10 s, 63 °C for 30 s then 72 °C for 30 s) immediately followed with 15 cycles (98 °C for 10 s and 72 °C for 30 s). PCR products from the second PCR were loaded onto a 2% gel and ran for 1 h at 90 V. Band appearing at ~240 bp was excised, quantified, mixed, and sequenced using a HiSeq 2500 sequencer (Illumina, California, USA). The sgRNA sequences against specific genes were recovered after removal of the tag sequences using Checkout [http://100bp.wordpress.com] and cutadapt (ver.1.12).

Enrichment of sgRNAs and genes was analyzed using MAGeCK (ver. 0.5.9) by comparing read counts from TE-671 cells after ExTxA selection with counts from cells without ExTxA selection to obtain a list of enriched genes[43]. $P < 0.01$ was considered statistically significant as defined by the Benjamini–Hochberg procedure[43]. Additionally, enrichment of sgRNAs and genes between the unselected control and the group treated without ExTxA (only veratridine and ouabain) was conducted to ensure no genes were enriched because of treatment with veratridine or ouabain.

### Membrane potential assay
HEK293-Na$_V$1.7 cells were transfected with TMEM233, CRELD1, LMAN2L, MMGT1, RNF121 and STT3B and co-transfected with five plasmids coding for the Glycosylphatidylinositol (GPI) complex (GPAA1, PIGK, PIGS, PIGU, PIGT) using Lipofectamine 2000. After 24 h, transfected cells were harvested and plated at a density of 10,000–15,000 cells per well on black-walled 384-well imaging plates. Plated cells were cultured for a further 24 h at 37 °C and 5% CO$_2$. FLIPR (Fluorescent Imaging Plate Reader) Red membrane potential assay dye (Molecular Devices, California, USA) was diluted in physiological salt solution according to the manufacturer's instructions (PSS, in mM: NaCl 140, glucose 11.5, HEPES 10, KCl 5.9, MgCl$_2$ 1.4, NaH$_2$PO$_4$ 1.2, NaHCO$_3$ 5, CaCl$_2$ 1.8, pH 7.4). Growth medium was removed from the plate and replaced with 20 μL of FLIPR Red membrane potential assay dye per well and incubated in the dark at 37 °C for 60 min. Changes in membrane potential were assessed using a FLIPR$^{Penta}$ (excitation, 515–545 nm; emission, 565–625 nm; Molecular Devices, California, USA) every 0.5 s for 10 min after the addition of ExTxA (10 μM). Fluorescence responses to ExTxA were computed as the maximum change in fluorescence over baseline ($\Delta F/F_0$) using ScreenWorks (Molecular Devices, Version 5.1.2.94).

### In vivo behavioral assessment
Nocifensive responses to intraplantar administration of ExTxA (5–10 nM) were assessed as previously described[38]. In brief, ExTxA was diluted in sterile saline/0.1% BSA and administered by an intraplantar injection (40 μL) into a single hind paw of Tmem233$^{Cre/Cre}$, Na$_V$1.7$^{Advill}$ and wild-type littermate controls. Pain behaviors, consisting of paw licks or flinches, were counted from video recordings by a blinded investigator for up to 60 min following injection.

### Calcium imaging
Dissociated DRG neurons from Tmem233$^{Cre/Cre}$ and C57BL/6 mice were plated on PDL-coated 96-well black-walled imaging plates and loaded with Fluo-4 AM calcium indicator (5 μM) (Invitrogen, Massachusetts,

USA) for 1 h in culture medium. Cells were washed with Hanks' balanced salt solution containing 20 mM HEPES and then transferred to the recording chamber of a Nikon Ti-E Deconvolution inverted microscope equipped with a Lumencor Spectra LED light source. Fluorescence images (excitation, 485 nm; emission, 521 nm) were acquired at 1 frame per second using a ×20 objective. Following the recording of baseline fluorescence (20 s), cells were consecutively exposed to buffer control (0.1% BSA, $t = 30$ s), ExTxA (10 nM, $t = 60$ s), and 30 mM KCl (30 mM, $t = 210$ s). ExTxA responders (>1.5-fold increase in fluorescence over baseline) were computed as a proportion of KCl-excitable cells and determined from a total of 4 cultures (2 WT, 2 Tmem233$^{Cre/Cre}$) yielding 1750 KCl-responsive neurons (899 WT, 851 Tmem233$^{Cre/Cre}$). Cells responding to buffer control were excluded from analysis.

### Proximity ligation assay (PLA)
HEK293 cells stably expressing β1/β2 were transiently transfected in T25 flasks using Lipofectamine 2000 with plasmids encoding GFP (0.35 μg) as a transfection control, as well as N-terminal HA-tagged TMEM233 (0.35 μg) and N-terminal FLAG- tagged Na$_V$1.7 (1.0 μg) alone or in combination. The total amount of plasmid DNA/T25 flask was kept constant at 1.7 μg flask using empty pcDNA3.1 plasmid as required. After 24 h incubation at 37 °C, cells were plated on PDL-coated glass coverslips and cultured for a further 24 h. Cells were washed once using sterile phosphate-buffered saline, fixed with 4% formaldehyde for 5 min, permeabilized with 0.1% Triton-X 100 in PSS, blocked using proximity ligation bocking buffer (Sigma DUO82007) and incubated for 1 h with anti-FLAG (1:1000; Sigma F1804) and anti-HA (1:1000; Sigma H6908) antibodies in Duolink antibody diluent. Permeabilized cells were then washed twice with wash buffer A at room temperature for 5 min. Anti-mouse PLUS (Sigma DUO92001) and anti-rabbit MINUS (Sigma DUO92005) probes were diluted 1:5 in Duolink antibody diluent and incubated with cells for 1 h at 37 °C. Cells were then washed with wash buffer A at room temperature for 5 min twice and incubated with ligase diluted 1:40 in ligation buffer. After incubation at 37 °C for 30 mins, cells were washed twice with wash buffer A. Polymerase was diluted 1:80 in amplification buffer and incubated with cells for 100 mins at 37 °C. Cells were subsequently washed with wash buffer B twice for 10 mins and mounted with Duolink In Situ mounting media with DAPI. Cells were imaged on a Zeiss AxioImage M2 with Apotome2 and Axiocam 506 camera. Proximity ligation assay signal (PLA particles) in GFP-positive cells were quantified using ImageJ in collapsed stacks from 3–6 random images obtained per coverslip (Image>Type>8-bit; Image>Adjust>Threshold; Image>Analyze>Analyze Particles). Particles size was set to 0.0003–∞ and circularity to 0.00–1.00. The average number of particles/GFP-positive cells from three independent experiments was analyzed by One-way ANOVA.

### Confocal microscopy
HEK293 cells were transfected to express TMEM233 using Lipofectamine 2000 (#11668019, ThermoFisher Scientific) as per manufacturer's instructions. Transfected HEK293 cells were then seeded within 6-well plates onto 13 mm diameter no 1/1.5 round coverslips 1 day before the experiment to achieve a target confluency of 50–70% at the time of treatment. On the day of the experiment, media was aspirated and replaced with biotinylated ExTxA (1 μM) diluted in fully complemented media followed by incubation at 37 °C/5% CO2 for 30 min. Media was then aspirated and cells were washed twice with pre-warmed media. Following incubation with streptavidin-Alexa488 conjugate (S32354, Themo Fisher Scientific Australia, 1:1000) for 30 min and a further two washes with complete media, cells were fixed for 11.5 min with 37 °C prewarmed 4% Electron microscopy grade Paraformaldehyde (#C004, ProSciTech) & 1x BRB80 (cytoskeletal stabilization buffer, 80 mM PIPES, 1 mM MgCl2, 1 mM EGTA, pH 6.8 with KOH). F-actin was stained with Rhodamine Phalloidin (Ab235138,

Abcam), and nuclei stained with 0.66 μg/mL Hoechst 33342 (#H3570, Thermofisher Scientific) diluted in antibody dilution buffer containing 150 mM NaCl, 20 mM Tris, 0.1% Triton X-100, 2% Bovine Serum Albumin and 0.1% NaN3, for 30 min. Cells were washed 5 times for 1 min each with PBS, mounted using 10% Mowiol/2.5% DABCO mounting media and imaged on an Andor Dragonfly spinning disc confocal.

HEK293 cells were transfected with N- and C-terminal HA-tagged TMEM233 with Lipofectamine 2000 (#11668019, ThermoFisher Scientific) as per manufacturer's instructions. 16 h after transfection, HEK293 cells expressing N- or C-terminally tagged TMEM233 were detached and seeded within 6-well plates onto 13 mm 1.5# coverslips. To determine cell surface exposure of N- or C-terminal tagged HA-TMEM233, 0.5 μg/mL of HA Tag monoclonal antibody (#26183, ThermoFisher Scientific) diluted in media was incubated with cells for 1 h inside a 5% CO2 tissue culture incubator with manual gentle agitation every 10–15 min, then fixed as per above. Control conditions were prepared by first seeding HEK293 cells transfected with N- or C-terminal HA-tagged TMEM233, then fixing, permeabilized and blocking as per above. Total N- or C terminal HA-tagged TMEM233 was detected using 0.5 μg/mL HA-tag antibody, diluted in antibody dilution buffer. For both conditions, the HA Tag was detected using a secondary anti-mouse antibody conjugated with alexafluor-488 (#715-545-150, Jackson ImmunoResearch Laboratories inc., 1:1000), nuclei were counterstained using 0.66 μg/mL Hoechst 33342.

### RNAscope in situ Hybridization (ISH)

Mouse Dorsal Root Ganglion (DRG) Frozen Sections (MF-240-C57, 7–10 μm thick) were either obtained commercially from Zyagen (www.zyagen.com) via AMS Biotechnology (https://www.amsbio.com) or produced in house. In the latter case adult C57BL/6 mice were deeply anesthetized with pentobarbital (i.p.) and transcardially perfused with heparinized saline (0.9% NaCl) followed by 25 mL of cold 4% paraformaldehyde in phosphate-buffered saline (pH 7.4). DRGs were extracted from the lumbar area and post-fixed with the same fixative solution for 2 h at 4 °C before being embedded in cryopreservative solution (30% sucrose) overnight at 4 °C. Tissue samples were then placed in OCT blocks for posterior sectioning by cryostat. 11 μm thick sections were mounted onto Superfrost Plus (Fisher Scientific) slides, allowed to freeze-dry overnight at −80 °C, for an immediate use, or were stored at −80 °C in air-tight containers for no longer than a month for subsequent experiments.

In situ hybridization was performed using the RNAscope assay (Advanced Cell Diagnostics, Bio-Techne) following the protocol for fresh-frozen samples, with 1 h post-fixing with 4% PFA in PBS at 4 °C and stepwise dehydration with 50%, 70 and 100% Ethanol. Tissue pre-treatment consisted of hydrogen peroxide and Protease IV (10 and 20 min, respectively) at RT. Following pre-treatment, probe hybridization and detection with the Multiplex Fluorescence Kit v2 were performed according to the manufacturer's protocol.

Probes included mmTmem233 (#519851-c1), mmScn9a (#3133341-c2) or mmScn9a (#3133341-c4), and mmNefh (#443671-c2) or mmNefh (#443671-c4). RNA localization was detected with either AF488 or Opal 520 (green), TS-coumarin TS405 (blue), Opal570 (red) and Opal 650 (far-red) fluorochrome dyes (Perkin Elmer) compared to DAPI staining (nuclei). ISH slides were mounted using Prolong Gold (ThermoFisher Scientific #P36930).

Fluorescence was detected using Zeiss LSM 880 Airyscan microscope by either using airyscan or LSM scanning. Images were taken at 10x and 20x magnifications with 4x averaging. Tiles were stitched when more than one was used to image the area, airyscan processed and exported as 16-bit uncompressed tiff files for further basic editing in Adobe Lightroom v6 (Adobe) on a color calibrated iMac (X-Rite) retina monitor. Final images were exported as jpeg files with 6000 pix on longest side at 300 ppi.

### Insecticidal assay in Drosophila melanogaster

To test for insecticidal activity, we injected ExTxA into three-day old *Drosophila melanogaster* females (average weight 0.875 mg). In total 77 *D. melanogaster* were used for this study: 52 individuals were each injected with 0.5 nmol (50 nL of a 0.1 mM solution) ExTxA, while 25 individuals served as control group and were injected with vehicle control (3% acetonitrile, 0.1% bovine serum albumin in water) and scored by a blinded observer.

### Co-purification and SDS-PAGE of Na$_V$1.7/TMEM233

The genes of human Na$_V$1.7 alternative splicing variant 3 (Uniprot accession: 15858-3) and TMEM233 (Uniprot accession: B4DJY2) were subcloned into a modified pEG BacMam vector. A green fluorescent protein (GFP) and a Twin-Strep tag were fused to the C-terminus of Na$_V$1.7, and an IL-2 signal peptide and a red fluorescent protein (mcherry) tag was fused to the N-terminus of TMEM233. Na$_V$1.7-GFP and mCherry-TMEM233 were co-expressed in HEK293F cells which were cultured in OPM-293 medium supplemented with 1% (v/v) fetal bovine serum (FBS, PAN-Biotech) in a 37 °C incubator with 5% CO$_2$ shaking at 100 rpm. The cells were collected and stored in a −80 °C freezer.

The cell pellets were resuspended in buffer A (20 mM HEPES, 150 mM NaCl, 2 mM β-mercaptoethanol (β-ME), pH 7.5, and protease inhibitor cocktail including 1 mM phenylmethyl-sulfonyl fluoride (PMSF), 0.8 μM pepstatin, 2 μM leupeptin, 2 μM aprotinin and 1 mM benzamidine). Then the membrane debris was collected by ultra-centrifugation and was resuspended in buffer A supplemented with 1% (w/v) n-Dodecyl-β-D-maltoside (DDM, Anatrace) and 0.15% (w/v) cholesteryl hemisuccinate (CHS, Anatrace). After ultra-centrifugation, the supernatant was loaded onto Streptactin Beads (Smart-Lifesciences). After washing, the protein complex was eluted in buffer A plus 0.05% GDN (w/v) and 5 mM desthiobiotin. The concentrated sample was loaded onto a Superose 6 Increase 10/300 GL column (GE healthcare, USA) pre-equilibrated with 20 mM HEPES, 150 mM NaCl, 0.007% GDN (w/v) and 2 mM β-ME, pH 7.5. Peak fractions were analyzed by HPLC with GFP, mCherry and UV detection and SDS-PAGE.

### Flow cytometry

HEK293 cells (untransfected controls or transfected with TMEM233) as well as control and TMEM233-transfected HEK293-Na$_V$1.7 cells were harvested using TrypLE Express, washed and resuspended in phosphate buffer solution. Cells were then pelleted (400 × g for 5 min) and re-suspended in physiological salt solution (PSS) containing 0.1% bovine serum albumin. Biotinylated ExTxA (1 μM) was then added to the cells and incubated on a shaker for 30 mins. Subsequently, cells were pelleted and washed three times with PSS. Cells were then incubated with Streptavidin DyLight™ 680 (1:2000) for 45 mins with agitation. Following, cells were washed three times with PSS and resuspended in PSS and analyzed using a BD FACSAria Cell Sorter. Flow cytometry data (100,000 events per group) were analyzed using FlowJo™ (version 10.7.2, FlowJo, LLC, Ashland, OR, USA). All gates were created on the unstained control and then applied to the other samples. FSC-A vs. SSC-A gating was used to select the cell population and to eliminate cell debris. Afterwards, doublet discrimination was performed using SSC-H vs. SSC-W and FSC-H vs. FSC-W gating. In the FSC-A vs. APC-A dot plot, a threshold was set at the fluorescent intensity of 1000 above which the events were considered as a positive signal.

### ExTxA-488 binding assay

HEK293 cells (untransfected controls, or transfected with TMEM233, Na$_V$1.7 or both TMEM233 and Na$_V$1.7), were incubated in suspension with ExTxA conjugated to Alexa Fluor 488 (ExTxA-488) in assay buffer (250 mM NaCl, 25 mM HEPES, pH 7) for 30 min at 4 °C. Unbound ExTxA-488 was removed by washing cells three times with assay buffer. Cells were plated at a density of 100,000 cells per well in a volume of 50 μL in a 384-well OptiPlate (PerkinElmer, Massachusetts, USA).

Fluorescence intensity (excitation 490 nm; emission 520 nm) was assessed using an Infinite M1000 Pro (Tecan, Männedorf, Switzerland). Concentration-response curves were fitted using the log (agonist) vs. response−variable slope (four parameters) equation and normalized to the largest fitted top from amongst the four groups as computed by GraphPad Prism Version 9.4.1.

## Data analysis

All data are presented as mean ± S.E.M. throughout unless otherwise stated. Details on n values (independent experiments or biological replicates) and statistical tests are detailed in Supplementary Table 1. All measurements were taken from distinct samples. Data were plotted and analyzed using GraphPad Prism version 9.4.1, as detailed throughout. Statistical significance was defined as $p < 0.05$.

## Reporting summary

Further information on research design is available in the Nature Portfolio Reporting Summary linked to this article.

# Data availability

The data that support this study are available from the corresponding authors upon request. Source data are provided as a Source Data file. Source data are provided with this paper.

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

## Acknowledgements

This work was supported by the University of Queensland Protein Production Facility and the Australian Cancer Research Foundation (ACRF) Cancer Biology Imaging Facility. Financial support was provided by the Australian Research Council (ARC; DP200102377, CE200100012) and the Australian National Health and Medical Research Council (NHMRC) fellowship to I.V. (APP1162503), Project grant (APP1162597) to M.M. and L.D.R., and Medical Research Council grant to J.J.C. and A.L.O. (MR/R011737/1). B.C.A. received funding from an NHMRC CJ Martin Fellowship (APP1162427). J.R.D. received funding through an ARC DECRA fellowship (DE210100422). S.D.H. and S.G.W. were supported by Merit Review Award B9253-C from the U.S. Dept. of Veterans Affairs Rehabilitation Service; The Center for Neuroscience & Regeneration Research is a Collaboration of the Paralyzed Veterans of America with Yale University. M.R.I. and D.A.A. received grants from the UK Medical Research Council (MR/S003428/1, MR/W002426/1). J.J.C., S.L., A.M.H., J.Zhao. and J.N.W. were supported by the Wellcome Trust (200183). A.M.H. receives funding from Qatar University (QUSD-CMED-2018/9-3; QUCG-CMED-19/20-4). We thank Dr. Patrick Walsh and Vincent Truong from Anatomic Inc. for helpful discussions on the gene expression profile of Anatomic Inc. iPSC-derived sensory neurons; Virginia Nink (Queensland Brain Institute, University of Queensland) for expert assistance with flow cytometry; and Prof Jenny Stow and Dr Lin Luo (Institute for Molecular Bioscience, University of Queensland) for provision of FANTOM Riken cDNA plasmids.

## Author contributions

S.J., J.R.D., T.K., X.C., P.Z., M.R.I., B.C.-A., L.D.R. performed electrophysiology experiments; S.J., J.R.D., T.K., S.L., V.S., and A.M.H. performed behavioral experiments; A.L.O., R.J.J., I.V., and S.D.R. performed imaging experiments; S.J., S.K., F.C., L.R. and Å.A. performed sequencing and molecular biology experiments; J.Zhang., Y.K.-Y.C., Y.Z., J.L., T.C., and S.P. performed biochemistry experiments; I.V. performed flow cytometry experiments; P.T., H.N.T.T. and K.M. synthesized compounds. D.A.A., L.D.R., J.N.W., J.Zhao., S.J.S., M.M., A.L., D.J., J.J.C., S.G.W., S.D.D., G.G.N., T.D., and I.V. designed the experiments, analyzed and interpreted the data, and edited the paper. T.D. and I.V. conceived the study and wrote the manuscript.

## Competing interests

The authors declare no competing interests.

## Additional information

¹Institute for Molecular Bioscience, The University of Queensland, St Lucia, QLD 4072, Australia. ²Department of Neurology and Center for Neuroscience and Regeneration Research, Yale University School of Medicine, New Haven, CT, USA. ³Rehabilitation Research Center, Veterans Affairs Connecticut Healthcare System, West Haven, CT, USA. ⁴Dr. John and Anne Chong Lab for Functional Genomics, Charles Perkins Centre, Centenary Institute, University of Sydney, Camperdown, NSW 2006, Australia. ⁵Molecular Nociception Group, Wolfson Institute for Biomedical Research, Division of Medicine, University College London, Gower Street, London WC1E 6BT, UK. ⁶Institute of Physics, Chinese Academy of Sciences, 100190 Beijing, P.R. China. ⁷Centre for Advanced Imaging, The University of Queensland, St Lucia, QLD 4072, Australia. ⁸School of Biomedical Sciences, The University of Queensland, St Lucia, QLD 4072, Australia. ⁹Wolfson Centre for Age-Related Diseases, Institute of Psychiatry, Psychology & Neuroscience, King's College London, SE1 1UL London, UK. ¹⁰College of Medicine, QU Health, Qatar University, PO Box 2713 Doha, Qatar. ¹¹Department of Anesthesiology and Intensive Care Medicine, Hannover Medical School, Hannover 30625, Germany. ¹²Australian Research Council Centre of Excellence for Innovations in Peptide and Protein Science, St Lucia, QLD 4072, Australia. ¹³School of Pharmacy, The University of Queensland, Woolloongabba, QLD 4102, Australia. ¹⁴These authors contributed equally: Sina Jami, Jennifer R. Deuis, Tabea Klasfauseweh. ✉e-mail: t.durek@imb.uq.edu.au; i.vetter@uq.edu.au

