## [Peer Review File · Nature Communications]

Pain-causing stinging nettle toxins target TMEM233 to modulate NaV1.7 functionReviewers' Comments:

Reviewer #1:

Remarks to the Author:

This manuscript describes remarkable findings that identify a previously unknown voltage-gated sodium channel subunit (TMEM233) that is revealed due to it being required for a response to a plant venom peptide toxin ExTxA (Excelsatoxin A). ExTxA has the remarkable property of blocking channel fast inactivation, leading to a persistent current after activation. The authors provide a wide range of functional and biochemical evidence that TMEM233 is both required for ExTxA function, and that TMEM233 interacts with the channel. Overall, the data are compelling and reveal a truly fascinating discovery.

TMEM233 is presented as a component of the Nav1.7 channel, but in fact it has similar functional consequences on all subtypes tested (Nav1.1-1.7). This is an important point, as it shows that TMEM233 must be interacting with a common Nav element. It is not clear why the authors

The data supporting the TMEM233 C-out topology would be aided with a cartoon.

The authors show that two related Dispanins (PRRT2 and TRARG1) are capable of binding to the channel (Fig. 3g). There is a weaker effect of both. It would be helpful to show a sequence alignment of the three proteins (TMEM233, PRRT2, and TRARG1) so that one can have a sense for how similar or different they are. Since they all are capable of endowing the channel with the ExTxA response, setting up the next question of how needs this information, even if the details are not addressed here.

Reviewer #2:

Remarks to the Author:

This is a fascinating paper. The authors have tracked down the mechanism by which an unusual plant-derived peptide toxin modifies voltage-dependent sodium channels in nociceptors to produce pain. and the exploration led them to the discovery of a previously-unknown accessory subunit of sodium channels, or at least a membrane protein that must be closely associated with sodium channels, a transmembrane protein called TMEM233. I do not think I have ever read a scientific paper with such an gripping detective-novel-like narrative, starting from the unexpected failure of the toxin to modify heterologously expressed channels even when co-expressed with every known beta subunit through to the convincing evidence that the toxin requires direct interaction with the TMEM233 protein to have its strong effect on sodium channel gating. The solution to the puzzle came from an CRISPR/Cas9 knockdown screen in one of the only cell lines found to reproduce the effect of the toxin, leading the authors to the previously obscure TMEM233 as the key player in the effect of the toxin on nociceptors. The work in the paper constitutes an amazing combination of experimental tools, including use of multiple knockout mice, electrophysiology with a wide range of native cells, cell lines, and huge number of combinations of heterologously-expressed proteins, biochemistry, high-resolution fluorescence microscopy. The work in the paper is so comprehensive I almost expected it to conclude with a cryo-EM structure of Nav1.7 channels in combination with TMEM233 proteins with the toxin bound.

Altogether, the paper is a remarkable achievement of scientific investigation. Although the huge amount of experimental data using a wide range of techniques makes the paper very dense, the figures are well-organized and as digestible as they could be. The major conclusion that TMEM233 is necessary for the potent action of ExTxA seemed convincing. Although it remains to be determined how the toxin binding to the channel-TMEM233 complex alters movement of the channel protein to change gating, the paper is complete as it stands, and understanding what residues in the alpha subunit are involved in the inhibition of inactivation by the TMEM233-toxin complex will likely require a lot of work. I did wonder whether it is possible that the binding of the toxin to the TMEM233 protein might serve more as mechanism to greatly increase the local concentration of the toxin near the

sodium channel rather than being absolutely necessary for toxin interaction with the channel...this could be tested by testing high concentrations of toxin, say 100 μ M, in the absence of the TMEM233. However, such experiments might require prohibitively large amounts of toxin. The possibility that there could be weak interactions of toxin with the channel in the absence could also be tested more completely by incubating the channels in the absence of TMEM233 for longer than the 5 minute exposure to 1 μ M toxin that was used...for example, doing a preincubation for an hour or two to see if there was any effect on the completeness of inactivation in the absence of TMEM233. However, such experiments are really to start to understand the molecular mechanisms involved in more detail, and the paper certainly does not need any more data to be compelling. The paper is well-written throughout, including in the very extensive Methods, and I could find only few places where the wording might be more precise.

line 112: " here the ExTxA-induced persistent current was inhibited by the selective NaV1.7 blocker Pn3a10 (100 nM) (Fig. 1 g,h). ExTxA-induced persistent TTX-sensitive currents in TE-671 neuroblastoma cells were also blocked by Pn3a, consistent with our previous observation that TE-671 cells express NaV1.7 as the predominant TTX-sensitive NaV subtype (Fig. 1i,j). In both cases, Pn3a does not completely abolish the ExTxA-induced persistent current, as might be expected if there is some contribution of other TTX-sensitive Navs in the effect. The wording might better be "largely inhibited" rather than "inhibited" and especially "blocked".

line 380: "TMEM233 had minor, though statistically significant, effects on the voltage-dependence of fast inactivation (-3.94 mV), but no effect on the voltage-dependence of activation (Fig. 5a-e). Small effects on the voltage-dependence of slow inactivation (-5.52 mV; Fig. 5f,g)... The statistical precision does not call for going to two decimal points in the numbers.

Response to reviewers

We thank the reviewers' for their positive comments.

We address specific suggestions below, and have highlighted corresponding amendments to the manuscript.

REVIEWERS' COMMENTS

Reviewer #1 (Remarks to the Author):

1. The data supporting the TMEM233 C-out topology would be aided with a cartoon.

Response: *We now include the AlphaFold2 structure prediction for TMEM233 in Supplementary Figure 4h to illustrate the TMEM233 topology more clearly.*

2. The authors show that two related Dispanins (PRRT2 and TRARG1) are capable of binding to the channel (Fig. 3g). There is a weaker effect of both. It would be helpful to show a sequence alignment of the three proteins (TMEM233, PRRT2, and TRARG1) so that one can have a sense for how similar or different they are. Since they all are capable of endowing the channel with the ExTxA response, setting up the next question of how needs this information, even if the details are not addressed here.

Response: *We now include a sequence alignment of human TMEM233, PRRT2 and TRARG1 in Supplementary Figure 4i.*

Supplementary Fig. S4. Effects of ExTxA on Nav1.7 co-expressed with the dispanins.

a) ^1H - ^{15}N transverse relaxation optimized spectroscopy (TROSY) spectra of ^{15}N labelled recombinant ExTxA in acetonitrile/ H_2O and dodecylphosphocholine micelles (DPC) showing distinctly different chemical shifts in a lipid membrane-like environment. **b-g)** Electrophysiological characterization of ExTxA effects on Nav1.7 co-expressed with

*TMEM233. b) Current-voltage (IV) relationship, c) conductance-voltage (GV, circles) and steady-state inactivation (squares) curves, d) V_{50} of activation and e) V_{50} of steady-state fast inactivation as well as f) time-dependence of recovery from fast inactivation and g) (left) peak ramp currents and (right) representative ramp (0.2 mV/s) current traces from HEK293 cells co-expressing Nav1.7 and TMEM233 after the addition of 0.1% BSA (yellow, control) and ExTxA (teal, 1 μ M in 0.1% BSA). h) AlphaFold2 TMEM233 (UniProt ID B4DJY2) structure prediction. Dotted lines indicate approximate locations of plasma membrane. i) Sequence alignment of the human dispanins TMEM233 (DSPB2; UniProt ID B4DJY2), PRRT2 (DSPB3; UniProt ID Q7Z6L0), and TRARG1 (DSPB1; UniProt ID Q8IXB3). The large N-terminal domains of PRRT2 and TRARG1 are truncated for clarity. j) Representative current traces from HEK293-Nav1.7 cells co-transfected with TMEM233, k) PRRT2 or l) TRARG1 showing ExTxA (1 μ M)-induced inhibition of inactivation. Shown is a 50 ms depolarization to -20 mV from a holding potential of -90 mV. Data are shown as mean \pm SEM; *, $p < 0.05$. n values and statistical information are detailed in Supplementary data Table 1.*

Reviewer #2 (Remarks to the Author):

1. I did wonder whether it is possible that the binding of the toxin to the TMEM233 protein might serve more as mechanism to greatly increase the local concentration of the toxin near the sodium channel rather than being absolutely necessary for toxin interaction with the channel...this could be tested by testing high concentrations of toxin, say 100 μ M, in the absence of the TMEM233. However, such experiments might require prohibitively large amounts of toxin. The possibility that there could be weak interactions of toxin with the channel in the absence could also be tested more completely by incubating the channels in the absence of TMEM233 for longer than the 5 minute exposure to 1 μ M toxin that was used...for example, doing a preincubation for an hour or two to see if there was any effect on the completeness of inactivation in the absence of TMEM233.

Response: *We thank the reviewer for raising this important point. Indeed, we agree that the most likely model is that the association of TMEM233 and ExTxA serves to enhance an otherwise weak, or very weak, interaction of the toxin with the channel. Unfortunately, we have observed no evidence of enhanced inactivation with low concentrations of toxin even over several days of incubation – for example during our initial attempts to establish our CRISPR screen assay – and testing higher concentrations of toxin is not only limited by availability of material, but also by toxin solubility and the high concentrations of organic solvents that would be required for these experiments. To address this point, we have expanded our discussion as follows:*

“It is likely that the TMEM233-ExTxA interaction serves to potentiate an otherwise low affinity toxin effect on $Na_v1.7$, albeit the molecular basis of this interaction remains to be determined.”

2. line 112: “ here the ExTxA-induced persistent current was inhibited by the selective $Na_v1.7$ blocker Pn3a10 (100 nM) (Fig. 1 g,h). ExTxA-induced persistent TTX-sensitive currents in TE-671 neuroblastoma cells were also blocked by Pn3a, consistent with our previous observation that TE-671 cells express $Na_v1.7$ as the predominant TTX-sensitive Na_v subtype (Fig. 1i,j). In both cases, Pn3a does not completely abolish the ExTxA-induced persistent current, as might be expected if there is some contribution of other TTX-sensitive Na_v s in the effect. The wording might better be “largely inhibited” rather than “inhibited” and especially “blocked”.

Response: *We have amended our wording to “largely inhibited” and “significantly reduced” as suggested.*

3. line 380: “TMEM233 had minor, though statistically significant, effects on the voltage-dependence of fast inactivation (-3.94 mV), but no effect on the voltage-dependence of activation (Fig. 5a-e). Small effects on the voltage-dependence of slow inactivation (-5.52 mV; Fig. 5f,g)... The statistical precision does not call for going to two decimal points in the numbers.

Response: *We have changed the reported values to include one decimal point.*